# Molecular basis for the catalytic mechanism of human neutral sphingomyelinases 1 (hSMPD2)

Jingbo Yi[1,4], Boya Qi[1,4], Jian Yin[1,4], Ruochong Li [1], Xudong Chen[1], Junhan Hu[1], Guohui Li [2], Sensen Zhang [1] ✉, Yuebin Zhang [2] ✉ & Maojun Yang [1,3] ✉

Enzymatic breakdown of sphingomyelin by sphingomyelinase (SMase) is the main source of the membrane lipids, ceramides, which are involved in many cellular physiological processes. However, the full-length structure of human neutral SMase has not been resolved; therefore, its catalytic mechanism remains unknown. Here, we resolve the structure of human full-length neutral SMase, sphingomyelinase 1 (SMPD2), which reveals that C-terminal trans-membrane helices contribute to dimeric architecture of hSMPD2 and that D111 − K116 loop domain is essential for substrate hydrolysis. Coupled with molecular docking, we clarify the binding pose of sphingomyelin, and site-directed mutagenesis further confirms key residues responsible for sphingo-myelin binding. Hybrid quantum mechanics/molecular mechanics (QM/MM) molecular dynamic (MD) simulations are utilized to elaborate the catalysis of hSMPD2 with the reported in vitro substrates, sphingomyelin and lyso-platelet activating fator (lyso-PAF). Our study provides mechanistic details that enhance our knowledge of lipid metabolism and may lead to an improved understanding of ceramide in disease and in cancer treatment.

Ceramide belongs to the second largest class of membrane lipids, sphingolipids, which participate in several important physiological processes such as apoptosis, senescence, cell growth, autophagy, and angoigenesis[1–3]. In response to extracellular stimuli, sphingo-myelinases (SMases) remove the phosphocholine head group from sphingomyelin[4] to rapidly generate ceramides. SMases are categor-ized into acid SMases (ASMases), neutral SMases (nSMases), and alkaline SMases according to their optimal catalytic pH, cation dependence, primary structure and cellular localization[5]. In mam-mals, nSMases play pivotal roles in ceramide-related signal trans-duction. There are four nSMases in mammals, including nSMase1 (*SMPD2*), nSMase2 (*SMPD3*), nSMase3 (*SMPD4*), and mitochondria-associated nSMase (MA-nSMase; *SMPD5*)[6–9]. SMPD3 is well-studied and has established roles in the stress-induced cellular response[10],

exosome biogenesis[11], cell growth and arrest[12], bone mineralization[13], and cancer pathogenesis[12].

*SMPD2* was the first mammalian nSMase to be identified and cloned[6]. SMPD2 mainly locates to the endoplasmic reticulum (ER)[14] and nuclear matrix[15], and it requires divalent cations (Mg[2+]) for activation[16]. Phosphorylation of SMPD2 by JNK signaling stimulates ceramide generation and apoptosis[17], whereas knockdown of SMPD2 suppressed the T cell receptor-induced apoptosis[18] and inhibited amyloid peptide induced ceramide generation[19]. SMPD2 may be rela-ted to diabetic kidney disease, as lower SMPD2 expression levels and decreased ceramide synthesis have been observed in patients with this condition[20]. Accordingly, in renal HK2 cells, inducing expression of SMPD2 increased cellular ceramide levels and promoted autophagy[21]. Moreover, SMPD2 can increase the ceramide:sphingomyelin ratio,

[1]Ministry of Education Key Laboratory of Protein Science, Tsinghua-Peking Joint Center for Life Sciences, Beijing Advanced Innovation Center for Structural Biology, School of Life Sciences, Tsinghua University, Beijing 100084, China. [2]State Key Laboratory of Molecular Reaction Dynamics, Dalian Institute of Chemical Physics, Chinese Academy of Sciences, Dalian, China. [3]Cryo-EM Facility Center, Southern University of Science & Technology, Shenzhen, China. [4]These authors contributed equally: Jingbo Yi, Boya Qi, Jian Yin. ✉e-mail: zhang.ss@phytovent.com; zhangyb@dicp.ac.cn; maojunyang@tsinghua.edu.cn

thereby suppressing hepatocellular carcinoma[22]. All these studies suggest that SMPD2 is related in ceramide generation and sphingomyelin hydrolysis. However, overexpression of SMPD2 has no obvious effect on CD95/Fas receptor-mediated ceramide production[23] and has minimal effect on ceramide level in MCF-7 cells[24], which further raises concerns about the role of SMPD2 in sphingomyelin hydrolysis. These seemingly controversial investigations suggests that the role of SMPD2 in ceramide generation and sphingomyelin metabolism might be limited to specific cell types and certain signaling pathways. By utilizing radio-labeling experiments, Sawai et al. discerned the role of SMPD2 as a lyso-PAF phospholipase C both in vitro and in vivo[25], rather as a sphingomyelinase. However, in mice, SMPD2 deficiency does not cause any disease related to lipid storage and no detectable changes were observed in sphingomyelin and lyso-PAF metabolism[26]. The regulation and cellular function of hSMPD2 is still under debate and warrants further investigations. Together, these findings demonstrate that SMPD2 plays a crucial role in maintaining cellular homeostasis and programmed cell death; therefore, a more clear understanding of the catalytic mechanism of SMPD2 is needed to increase our knowledge of structural lipid metabolism pathways and their influence on cell physiology.

In this study, we solve the structure of full-length human SMPD2 (hSMPD2), which guides us to further explore the roles of TMD helices in maintaining the dimeric architecture, as well as the essential roles of the D111 – K116 loop domain in enzyme hydrolysis activity. We confirm key residues responsible for sphingomyelin binding to hSMPD2 and apply theoretical modeling approaches to elaborate the molecular underpinnings of sphingomyelin and lyso-PAF catalysis, which further establish the essential role of residue K116 and H272 in facilitating the catalytic process. Our structural, enzymatic, and theoretical investigations would provide deeper understanding of SMases family.

## Results

### Biochemical characterization of hSMPD2 enzyme activity

To better elucidate the biochemical properties of hSMPD2, we cloned Flag-tagged hSMPD2 into the pcDNA3.1(-) vector and transfected into the SY5Y cells. BODIPY™ FL labeled sphingomyelin was utilized to trace the localization of sphingomyelin. The overexpressed hSMPD2 was mainly localized to the ER and partially to the Golgi and the plasma membrane (Supplementary Fig. 1). Fluorescence co-localization between BODIPY™ SM and overexpressed hSMPD2 were observed on the Golgi, plasma membrane, and ER (Supplementary Fig. 1). Next, we overexpressed hSMPD2 in HEK293F cells and used affinity chromatography and related biochemical approaches to purify hSMPD2 for structural and functional studies (Supplementary Fig. 2). For a control, we separately cloned, expressed, and isolated the well-characterized hSMPD3 using the aforementioned approaches. We monitored hSMPD2 activity in vitro using the Amplex Red Sphingomyelinase Assay Kit and observed a robust increase in the fluorescence signal within the first fifty minutes of the reaction, indicating that hSMPD2 hydrolyzed sphingomyelin (Fig. 1a, b). Also, the hydrolysis activity of hSMPD2 was comparable to that of hSMPD3 (Fig. 1b), which was consistent with a previous study[27]. Since SMPD2 could functions as a lysophospholipase by hydrolyzing lyso-PAF as substrate[25], we also explored the enzymatic kinetic parameters of in vitro purified hSMPD2 (Supplementary Fig. 3). The $Km$ value for SM was comparable to that of lyso-PAF (Supplementary Fig. 3), in agreement with the previous report[25].

### Structure determination and overall structure of the hSMPD2

Purified hSMPD2 proteins were subjected to cryo-electron microscopy (cryo-EM) study, and we ultimately obtained a 3.07-Å hSMPD2 structure after sample preparation, images collection and data processing (Fig. 1c and Supplementary Figs. 1e–f, 4, 5 and Table 1). In contrast to previously reported monomeric SMase structures[28–32], hSMPD2 adopts

a dimeric architecture with rigorous binary symmetry and with four transmembrane helices (TMHs) protruding into the membrane (Fig. 1c). In contrast to the density for the TMHs, the densities for the loop preceding TMH1 and for residues 403 – 423 following TMH2 were disordered and were not modeled. hSMPD2 is composed of a transmembrane domain (TMD) and a catalytic domain (CAD) and occupies a spatial volume of approximately $85 \times 60 \times 80 \, \text{Å}^3$ (Fig. 1c, d). Each monomer mainly comprises two TMHs, twelve β-sheets, and four α-helices, forming a special L-shaped architecture (Fig. 1e–f). Sequence alignment between hSMPD2 and hSMPD3 showed low (22.3%) sequence identity between the two enzymes (Supplementary Fig. 6), although their catalytic domains share similar topology and critical residues located in the hydrophobic groove are conserved.

### Dimer interface of hSMPD2

The dimer interface of hSMPD2 is formed mainly by TMH2 within the TMD and by β6 and α3 within the CAD (Fig. 2a). The TMD of dimeric hSMPD2 consists of two pairs of symmetrical helices that stabilize the dimerization through direct interactions and that anchor the protein into the membrane to hydrolyze sphingomyelin. Tomiuk et al. found that removing the TMHs of hSMPD2 resulted in complete loss of enzymatic activity, indicating that the TMD plays a pivotal role in hSMPD2 localization and function[33]. Dimeric hSMPD2 is stabilized by substantial hydrophobic interactions and π – π interactions between TMH2 from each protomer. In particular, the T359 residues of TMH2 on each protomer interact with each other, and the W367 residues of TMH2 on each protomer form π – π interactions (Fig. 2b). In addition, residue R395 within TMH2 of one protomer interacts with Q91 within the CAD of the other protomer, and residues H164 and K168 from α3 of one protomer interact with E92 from β6 of the neighboring protomer (Fig. 2c). In the TMD interface, we also observed densities within the hydrophobic cleft, which we attributed to four acyl chains that interact extensively with residues F371, L373, and F374 within TMH2 and with residues P104, Y105, W112, and F113 within the CAD of both protomers (Fig. 2b and Supplementary Figs. 7a–b). The presence of these hydrophobic lipids contributes to the stabilization of dimeric hSMPD2. The hydrophobic lipids protrude into the dimer interface cleft and serve as a joint to interact with the TMD and the CAD.

Seeking experimental validation for the role of these residues in dimer formation and sphingomyelin hydrolysis, we constructed Q91A, E92A, Y105A, W112A, F113A, L356A, W367A, and T359A mutants of hSMPD2 with Strep tag and Flag tag. First, in vitro pull-down assays were performed with extracts from cells co-expressing Strep-tagged hSMPD2 and Flag-tagged hSMPD2 by using Strep-tagged proteins as "bait" protein. As illustrated in Fig. 2d, E92A mutants largely reduced the protein expression level compared with the wildtype, resulting in a decreased dimer stability. Multiple unbiased MD simulations (5 × 50 ns) after the equilibration phase indicates that E92 and K168 could form a stable salt bridge throughout the MD simulations, further verifying the essential role of this salt bridge interaction in dimer formation (Fig. 2e). Second, we purified mutant proteins as aforementioned to test in the enzymatic assay[34]. Q91A, E92A, L356A, W367A, and T359A mutants had similar sphingomyelin hydrolysis activity compared to wild type hSMPD2 (Fig. 2f and Supplementary Fig. 8), whereas Y105A, W112A, and F113A point mutations in hSMPD2, which are within the loop region of the CAD, robustly diminished sphingomyelin hydrolysis compared to wild type enzyme (Fig. 2f and Supplementary Fig. 8), indicating that mutations in these residues might affect hSMPD2 activity.

### hSMPD2 substrate-binding pocket

With the exception of nSMase3, all of the nSMases possess a DNase-I-type catalytic core[35], suggesting a common catalytic mechanism among nSMases. In our current hSMPD2 structure, we observed a magnesium ion located in a negatively charged cleft in the CAD that is

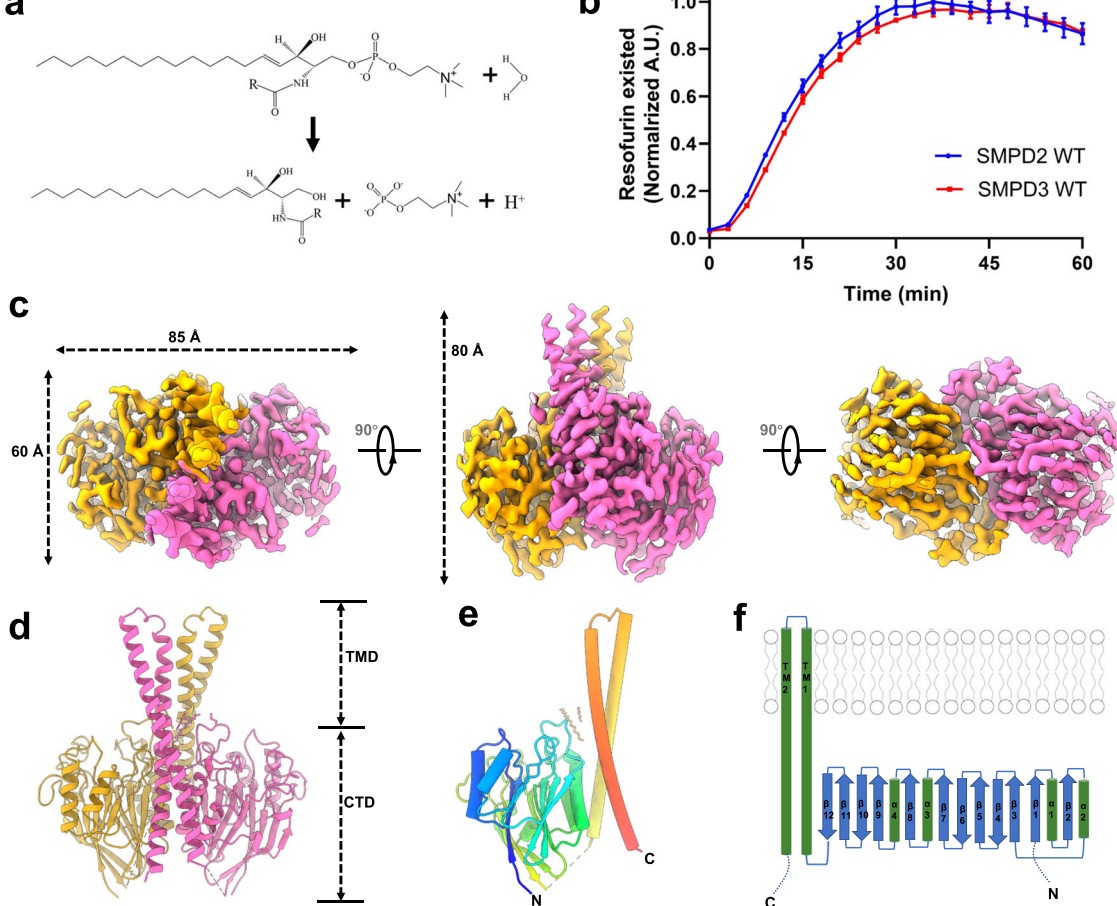

**Fig. 1 | Structure of hSMPD2. a** Hydrolysis of sphingomyelin by sphingomyelinases. **b** Relative enzyme activity of wild type hSMPD2 and hSMPD3 measured using the Amplex Red Sphingomyelinase Assay Kit. Data shown are mean +/− SEM ($n = 4$, biological replicates). Source data are provided as a Source Data file. **c** Cryo-EM map of the high-resolution hSMPD2 structure from three different views. The two protomers are indicated in orange and purple. **d** Ribbon representation of hSMPD2 from a side view. The two protomers are indicated in orange and purple. **e** Cylindrical representation of a hSMPD2 protomer. Different regions are indicated with different colors. **f** Schematic representation of the secondary structure of hSMPD2.

further coordinated by residues N15 and E49 in the active site (Fig. 3a and Supplementary Fig. 7c). Residues N15 and E49 are conserved among nSMases from bacteria to mammals (Supplementary Fig. 6). To verify the role of N15 and E49 in catalysis, we evaluated N15A and E49A single mutants of hSMPD2 in the enzymatic assay. Both the N15A and E49A mutations in hSMPD2 nearly eliminated sphingomyelin hydrolysis activity (Fig. 3b and Supplementary Fig. 8), which is consistent with a previous report that these mutations eliminated catalytic activity in Isc1p[36]. These results reveal the pivotal role for N15 and E49 in hSMPD2 sphingomyelin hydrolysis and indicate that Mg²⁺ is required for the reaction.

### D-K loop domain in hSMPD2

In the vicinity of the substrate binding groove in our hSMPD2 structure, residues D111 − K116 form a short helix that was named a 'P-loop-like domain' in Isc1p[36], and was subsequently renamed as 'DK switch' after the first crystal structure of the hSMPD3 catalytic domain was resolved in 2017[28]. Based on the acquired bacterial SMase structures, Airola et al. speculated a regulatory role for the DK switch[28], wherein the DK switch serves as an obstacle that impedes substrate approach. Once activated, the conserved residues D111 and K116 interact through a salt-bridge interaction, resulting in formation of a helix that distances the loop from the hydrophobic groove. The finding that the DK switch of hSMPD3 and bacterial SMase takes on several conformations may be caused by the fact that bacterial SMase is a monomer and the reported

hSMPD3 is a truncated structure and thus may lack sufficient interactions to fix the loop in a specific conformation.

The aforementioned Y105, W112, and F113 residues for which we confirmed an essential function in hSMPD2 catalytic activity are located along the D111-K116 loop domain, prompting us to further investigate the role of this loop region. The D111-K116 loop domain is strategically positioned near TMH2 and hydrophobic lipids in the dimeric interface, as well as near the helix α3 and the substrate binding pocket in the CAD (Fig. 3c). For instance, residues W112 and F113 within the D-K loop domain interact with the hydrophobic lipid from the cleft. Also, residues Y105, Y103, and H109, which are within just six residues of the D111-K116 loop, interact with residues F374, H375, and E378 within TMH2 and with residues H151 and Y148 within helix α3 of the CAD (Fig. 3c).

We next generated several intramolecular interaction-related mutants and measured their enzyme activities. We found that K116A mutation in hSMPD2 completely eliminated the catalytic activity and D111A mutation achieved only 1% of the catalytic activity of wild type hSMPD2 (Fig. 3d), findings consistent with a previous study of mutations in these conserved residues in Isc1p[36]. In addition, Y103A and H109A mutants were not expressed as efficiently and were characterized by dramatically attenuated catalytic activity compared with wide type protein, whereas a S114A mutant was expressed and hydrolyzed sphingomyelin similarly to wild type protein (Fig. 3d and Supplementary Fig. 8). These functional findings were consistent with our structural analysis. Taken together, our results illustrate that the highly

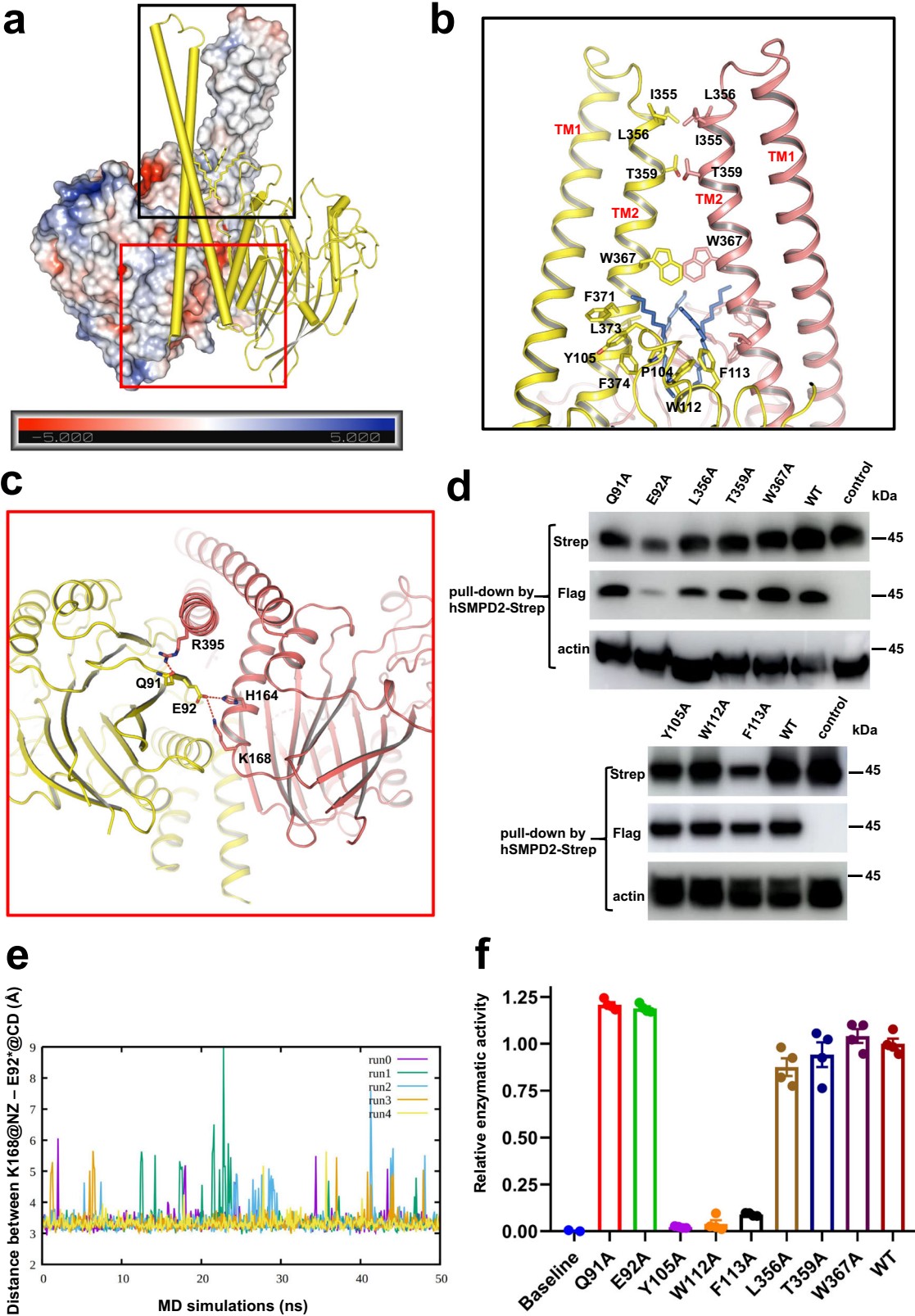

**Fig. 2 | Dimer interface of hSMPD2. a** Ribbon representation of hSMPD2 with one protomer shown in cylinder and the other protomer shown in surface electrostatics. **b** Hydrophobic interactions between the TM regions of hSMPD2. The four lipids are colored blue. **c** Magnification of the polar interactions between different monomers of hSMPD2 from the bottom view. **d** In vitro pull-down assays of the hSMPD2 mutants by using the Strep tagged hSMPD2 as a bait protein. Source data are provided as a Source Data file. **e** Multiple unbiased MD simulations after the equilibration phase indicates a stable salt bridge that formed by E92 and K168 throughout the MD simulations. **f** Relative enzyme activity of wild type hSMPD2 and the indicated dimer interface-related mutants; mean ± SEM, n = 4 biological replicates. Source data are provided as a Source Data file.

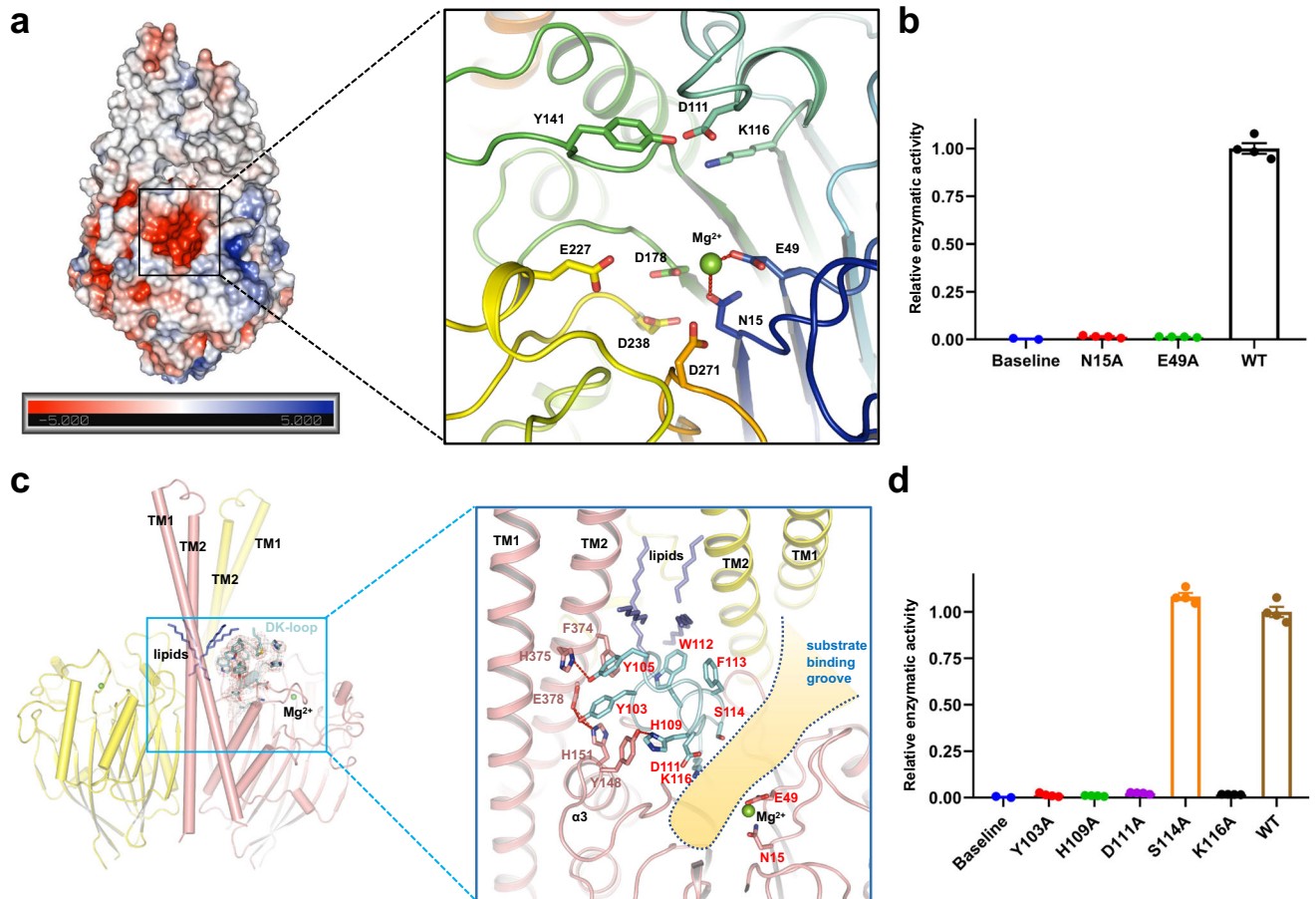

**Fig. 3 | Substrate-binding pocket and D-K loop in hSMPD2. a** Heatmap of the surface electrostatics of hSMPD2. Magnification of the Mg²⁺ and charged residues in the binding pocket of hSMPD2 (right). **b** Relative enzyme activity of wild type hSMPD2 and the indicated binding pocket-related mutants; mean ± SEM, $n = 4$ biological replicates. Source data are provided as a Source Data file. **c** Cylinder representation of hSMPD2 with the DK-loop regions were shown with side chain and superimposed into the cryo-EM density (left). The magnified view of the polar interactions between the DK loop and the neighboring residues (right). **d** The relative enzyme activity of WT and the DK loop related mutants in hSMPD2; mean ± SEM, $n = 4$ biological replicates. Source data are provided as a Source Data file.

conserved residues D111 and K116 are indispensable for sphingomyelin hydrolysis and suggest that mutations in the D111-K116 loop domain may disrupt essential interactions between the TMD and CAD that could impede the approach of substrate to the catalytic center and hinder hydrolysis.

## Molecular docking and molecular dynamics simulations

To further investigate the catalytic mechanism by which SMPD2 hydrolyses sphingomyelin into ceramide and phosphocholine, we next tried to solve the structures of hSMPD2 in different states by adding ceramide or sphingomyelin before cryo-sample preparation. However, we failed to obtain these states (the ceramide-bound and the sphingomyelin-bound structures). Thus, in silico molecular docking and molecular dynamic simulations were employed here to explore the binding orientation of sphingomyelin at the active site and along the hydrophobic groove. The initial conformation of sphingomyelin was manually placed along the hydrophobic groove of hSMPD2 and a library of 300 conformations of sphingomyelin (Supplementary Fig. 9a) was used to conduct the molecular docking according to the RosettaLigand docking protocol. The pose with a minimum distance (2.1 Å) between the magnesium ion and the non-bridging oxygen atom of the phosphate group of sphingomyelin among 600 docking conformations was selected as the possible binding mode (Fig. 4a and Supplementary Fig. 9b). Then, the dimeric architecture of hSMPD2 binding with the docked sphingomyelin was used as the initial

conformation to construct the molecular dynamics simulations system in the explicit membrane environment composed of a SM/POPS/POPC bilayer. During the initial equilibration phase (~5 ns) of our MD simulations, we observed that the side chain of K116 at one of the protomers exhibited a noticeable rearrangement to form the lysine-phosphate salt bridge with the binding sphingomyelin (Fig. 4b). Multiple unbiased MD simulations (5 × 50 ns) after the equilibration phase revealed the lysine-phosphate salt bridge would be well maintained during the production run phase, further indicating the amine group of K116 plays an essential role in stabilizing the scissile phosphate at the catalytic center (Fig. 4c). The single Mg²⁺ ion was directly coordinated by N15, E48, E49 and D178 and the interaction network was stably maintained throughout the classical MD simulations. In addition to the nucleophilic water that was located in front of the scissile phosphodiester bond, we noted another water molecule bridging the nucleophilic water and H272 via hydrogen-bonding interactions (Fig. 4a).

Before proceeding with theoretical investigations of the hydrolysis mechanism, we performed mutagenesis experiments to verify the reactant conformation. The docked pose indicated that the aliphatic chain of sphingomyelin is hydrophobically stabilized by I19 and W51 while the phosphocholine methyl groups of sphingomyelin would interact with W17. Site-directed mutagenesis shown that both W17A and W51A mutants completely eliminated the catalytic activity of hSMPD2, and that an I19A mutant retained only ~20% of enzymatic activity compared to wild type, suggesting that W17A, I19A, and W51A

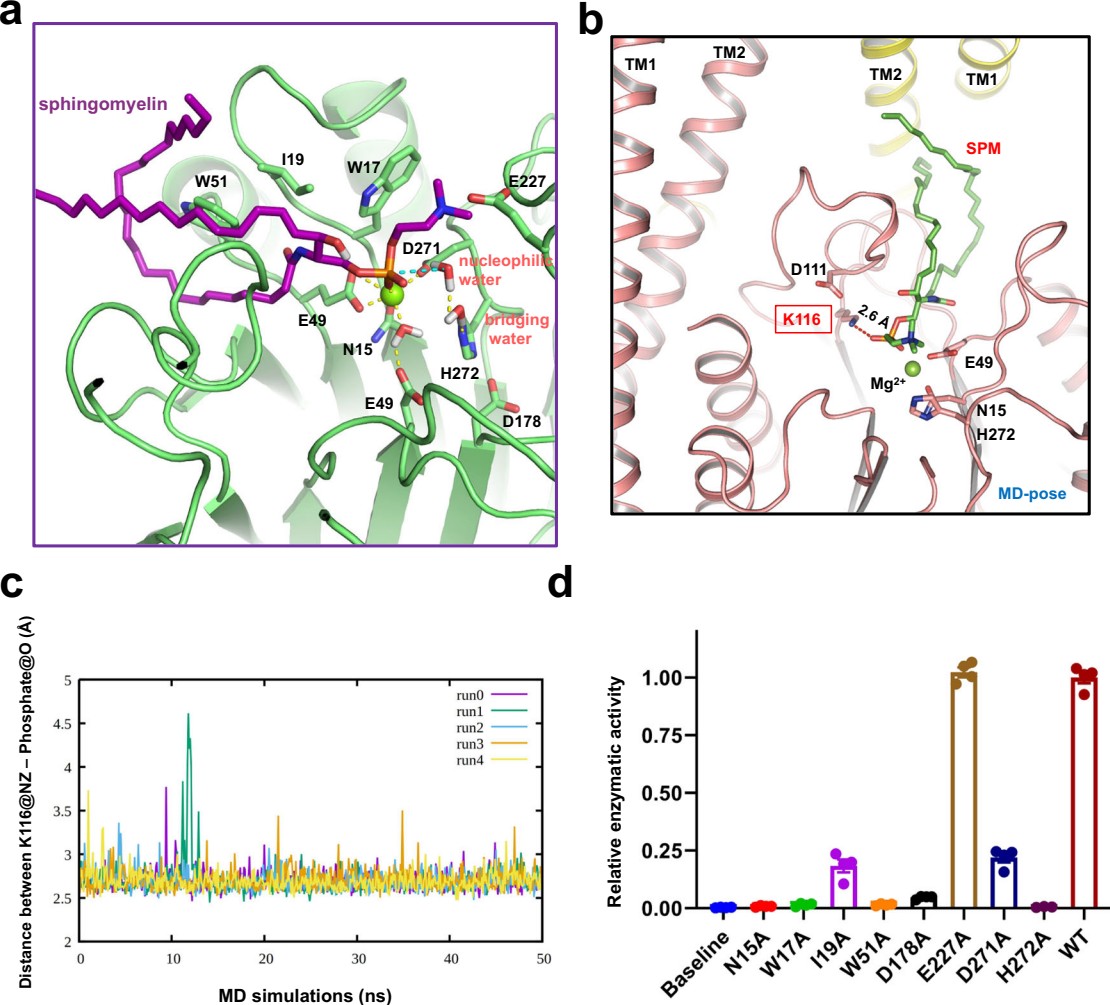

**Fig. 4 | MD simulations of the sphingomyelin binding within hSMPD2. a** MD simulations reveal the binding pose of sphingomyelin within the binding pocket. The magnesium ion is shown as a green sphere. **b** A representative MD pose reveals that the side chain of K116 could form a salt bridge with the phosphate group of the binding sphingomyelin. **c** Multiple unbiased MD simulations revealed the lysine-phosphate salt bridge would be well maintained during the production run phase. **d** Relative enzyme activity of wild type hSMPD2 and the indicated sphingomyelin-binding-pocket-related mutants; mean ± SEM, $n = 4$ biological replicates. Source data are provided as a Source Data file.

mutations impaired the binding of sphingomyelin (Fig. 4d and Supplementary Fig. 8) and impeded the catalytic process. To further probe the enzymatic kinetic parameters of hSMPD2, we separately measured representative point mutants, which showed that the I19A mutant increased the Km value (15.46 µM) of hSMPD2 by catalyzing sphingomyelin compared with WT proteins (Km = 3.0 µM), whereas the E49A mutant could not be fitted with an appropriate Km value due to its low catalytic rate (Supplementary Fig. 2). Furthermore, when we mutated residues that our proposed model indicates coordinating with $Mg^{2+}$ in the active site (N15A, E49A, D178A), hSMPD2 catalytic activity was abolished in each case (Fig. 4d, and Supplementary Fig. 8), indicating that precise $Mg^{2+}$ coordination is crucial for hydrolysis. Finally, we determined that the H272A mutant would also lead to the elimination of the catalytic activity of hSMPD2 (Fig. 4d and Supplementary Fig. 8). The key role of H272 in facilitating the catalytic process was explored in our following QM/MM free energy calculations.

## Sphingomyelin and Lyso-PAF hydrolysis proposed by QM/MM simulations

With extensive functional analysis of point mutants to support our model, we next performed QM/MM steered MD (SMD) simulations to drive the catalytic reaction in the forward direction from the

reactant state to the product state by defining the reaction coordinate as the distance difference (dd) between the forming bond and the scissile bond (Fig. 5a–c). The QM/MM SMD simulations were performed for 2.5 ps with the boundary of dd from −2 Å to 2.4 Å using the PBE0 functional together with the def2-SVP basis set. Then, umbrella sampling (US) was conducted to estimate the free energy profile along the reaction coordinate, in which 45 bins from −2 Å to 2.4 Å with 0.1-Å increments was used. The initial conformation of each bin was extracted from the QM/MM SMD simulation trajectory and 10 ps US QM/MM simulations of each bin were performed using the same PBE0/def2-SVP method. The last 3 ps trajectories were used for free energy profile estimation. By calculating the weights from US simulation, the 1D free energy profile could be mapped onto the 2D free energy surface which clearly shows the concerted nature of a typical SN2 reaction mechanism as shown in Fig. 5d. The proposed catalytic pathway will overcome a free energy barrier of 14.58 kcal/mol at dd = 0.36 Å, which corresponds to the deprotonation process of the nucleophilic water and formation of the hydroxyl group. The bridging water molecule accepts the leaving proton from the nucleophilic water and simultaneously releases one of its protons to H272; therefore, H272 acts as a general base to accept the proton and to facilitate the reaction

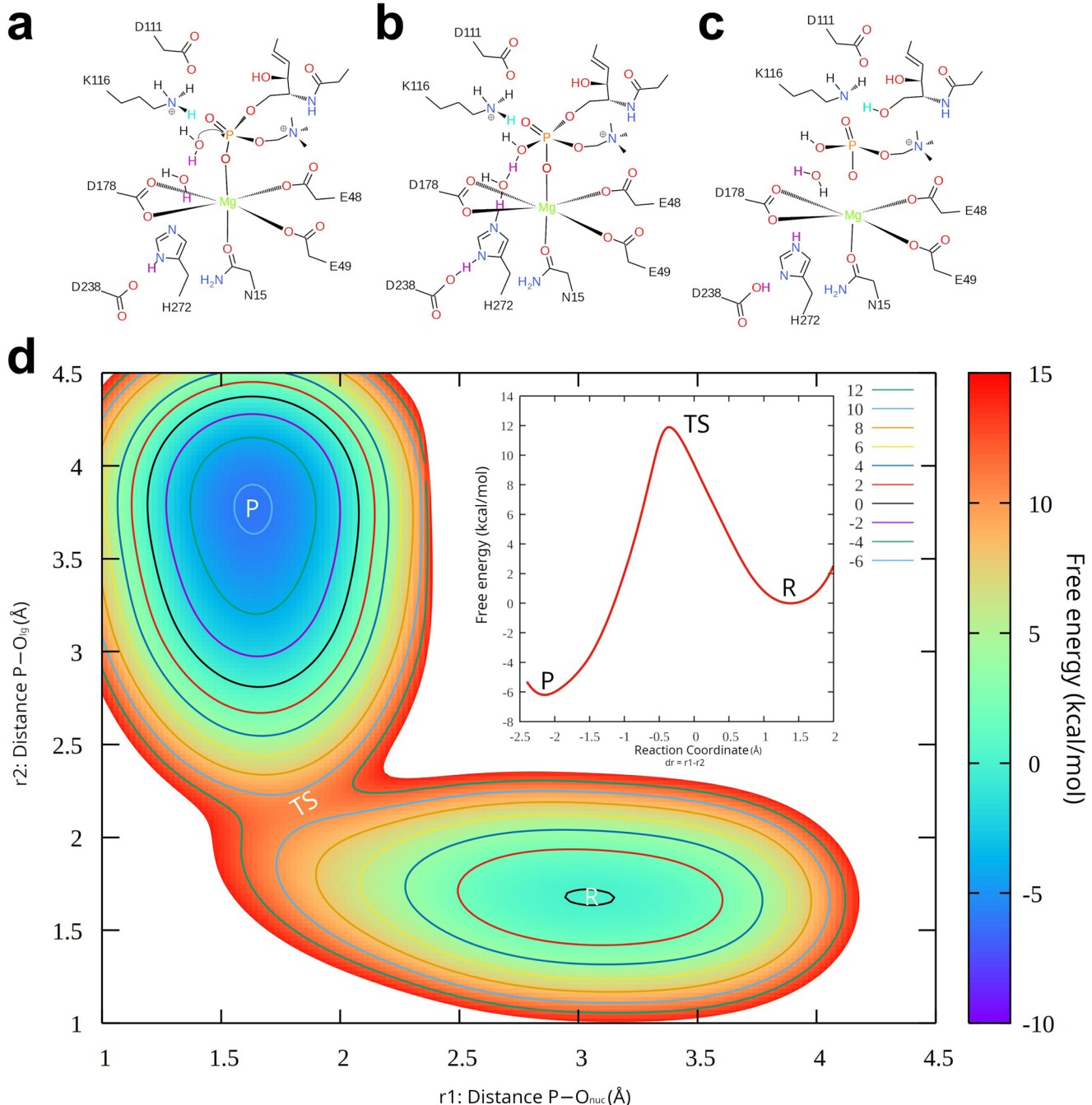

**Fig. 5 | Schematic representation of the hypothetical pathway for the hydrolysis reaction. a** the reactant (R); (**b**) transition state (TS); (**c**) product (P) state along phosphodiester bond cleavage; The protons transferred during the reaction are colored with purple. K116 provides a proton (colored with cyan) to stabilize the P state; (**d**) free energy surface along the forming P-$O_{nuc}$ bond and the leaving group P-$O_{lg}$ bond. The insert indicates the free energy profile as a reaction coordinate of dr = r1-r2.

and D238 will further serve as a proton reservoir to lower the proton transfer free energy barrier and maintain a neutral imidazole group of H272. After the inversion of the stereo configuration of the pentacovalent phosphorane intermediate and, then, the product state is formed with delG = −3.46 kcal/mol, suggesting the reaction process is an exergonic SN2 reaction. It should be noted that after the scissile phosphodiester bond is broken, K116 will transfer one of its proton to the leaving oxygen in stabilizing the final product state. Our unbiased MD simulations indicated that a stable salt-bridge was formed between D111 and K116 during five independent MD trajectories, indicating that D111 is indispensable in catalysis process (Supplementary Fig. 10).

In addition to probing the catalytic process of sphingomyelin, our current structure also offers the possibility to study other candidate substrates, such as lyso-PAF, which was investigated both in vitro and in vivo[25]. A parallel comparison study using the lyso-PAF as substrate was also conducted via QM/MM US simulations (Supplementary Figs. 11-12). The free energy profile for lyso-PAF along our proposed catalytic pathway demonstrated a higher free energy barrier compared with sphingomyelin and the hydrolysis reaction for lyso-PAF is an endergonic process with G = 2.34 kca/mol. According to our calculated free energy profiles, both the catalytic reaction would occur at room temperature although the lyso-PAF exhibits a higher activation barrier (Supplementary Fig. 12).

## Structure divergence among nSMases

Our hSMPD2 model represents the first for a mammalian full-length nSMase, and it is analogous to models reported for other nSMases[28–31], but with striking differences. The prokaryotic nSMases adopt an extra β-hairpin structure (Supplementary Fig. 13) that participates in binding to the membrane-bound sphingomyelin substrate[30], which is not present in mammalian nSMases. hSMPD2 adopts a dimeric architecture by utilizing the two C-terminal transmembrane helices (Supplementary Fig. 13), whereas ASMases has been proposed to adopt a dimeric structure through the N-terminal saposin domains[37] (Supplementary Fig. 13), all of which differ compared to the reported monomeric prokaryotic nSMases.

hSMPD2 and yeast Isc1p share 27.81% sequence identity and possess the similar topological architecture. To further investigate the species differences, we compared the structures of hSMPD2 with Isc1p (AlphaFold predicted) (Fig. 6a), and the results showed that the Root Mean Square Deviation (RMSD) of the two proteins was 1.353 Å over 246 Cα atoms (Fig. 6b). Notably, residues in the catalytic domain of the two proteins are identical, including the magnesium ion chelating residues (E49 and N15 from hSMDP2), catalytic reaction residues (H136, H272, and D178 from hSMPD2), and hydrophobic phospholipid stabilizing residues (W51 and W17 from hSMPD2). In the D-K switch region, D111, K116 and W112 of

hSMPD2 have a similar conformation to those in Isc1p, but other aromatic amino acids such as Y103, Y105 and F113 are not conserved between these two proteins. Although Isc1p also contains two TMHs at the C-terminus, the TMH sequences of these two proteins are not conserved, and the TMHs from Isc1p adopt a 10-degree shift deviation (Fig. 6b). The discrepancy in the TMHs and D-K switch residues may reflect evolutionary differences between species.

In addition, structural comparison between SMPD2 and SMPD3 showed that the RMSD of the two proteins was 3.567 Å over 138 Cα atoms (Fig. 6c). Both proteins have highly conserved magnesium ion chelating residues and catalytic reaction residues, but the phospholipid stabilizing residues (W17 and W51 from hSMPD2) are different. Specifically, the D-K switch structural domain of hSMPD3 is significantly different. Compared with hSMPD2, while the conformation of residue lysine remained essentially unchanged, D430 of hSMPD3 was spatially flipped upward, resulting in a significant change in the conformation of the D-K switch (Fig. 6c). At the same time, L432 and the adjacent L138 were in close proximity to each other, thus impeding the substrate binding process of sphingomyelin (Fig. 6d–f). Akin to the reported homologous nSMase structures, the current hSMPD2 structure represents a substrate binding groove exposed state, that is ready to bind and hydrolysis substrates.

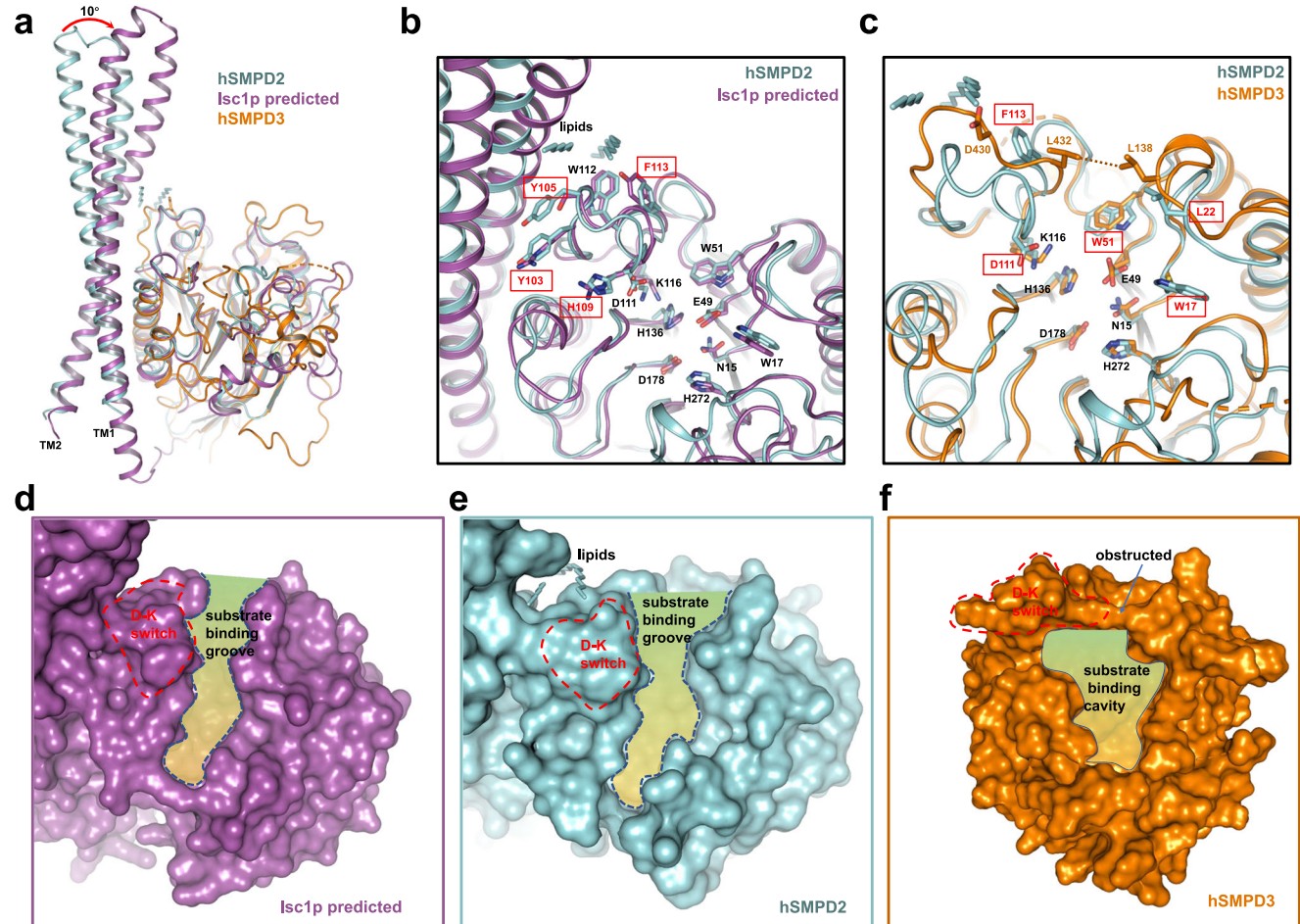

**Fig. 6 | Structural comparisons of nSMases. a** Overall structural comparison of monomeric hSMPD2 (cyan), hSMPD3 (PDB code 5UVG; orange) and CODE (alpha fold predicted; purple). The TMHs from Isc1p adopt a 10-degree shift deviation compared with that of hSMPD2. **b** Structural comparison of monomeric hSMPD2 (cyan) and Isc1p (alpha fold predicted; purple). The black labeled and red labeled residues indicates the conserved and variable residues within the binding pocket. **c** Structural comparison of monomeric hSMPD2 (cyan) and hSMPD3 (PDB code

5UVG; orange). The black label indicates the conserved residues between SMPD2 and SMPD3. The red and orange labels indicate the variable residues in hSMPD2 and hSMPD3, respectively. **d** Surface representation of the substrate binding groove in Isc1p (purple). **e** Surface representation of the substrate binding groove in hSMPD2 (cyan). **f** Surface representation of the obstructed substrate binding groove in hSMPD3 (orange).

## Discussion

In this study, we delineated the full-length structure of a human nSMase. Unlike the previously reported structures of SMases[28–31], hSMPD2 is a dimeric protein with four TMHs integrating into the membrane (Supplementary Fig. 14). Several lipids were observed within the dimeric hydrophobic cleft that participate in protein stabilization and interact with the D111-K116 loop domain. Different kinds of cell stress[18, 19, 38, 39] have been reported to stimulate the activity of SMPD2 (Supplementary Fig. 14). We completed a theoretical investigation to elucidate the binding pose and performed QM/MM MD simulations of hSMPD2 catalytic process of the reported substrates—sphingomyelin and lyso-PAF. The catalytic mechanisms of acid sphingomyelinase with two zinc ions observed at the catalytic center were proposed by previous studies[37,40], which are different from our current single-metal-mediated catalytic process (Fig. 5). The highly conserved residues N15, E49, D111, and K116 cooperate to form the catalytic core and are assisted by H272 (Supplementary Fig. 14). Our simulation studies revealed that amine group of K116 from the conserved D-K loop region plays an essential role in stabilizing the scissile phosphate at the catalytic center. A nucleophilic water releases its proton to form the hydroxyl group and a bridging water molecule facilitates the proton transfer (Supplementary Fig. 14). Subsequently, the hydroxyl group of ceramide participates in stabilizing the leaving oxygen after the scissile phosphodiester bond is broken, which in turn triggers cellular processes such as apoptosis (Supplementary Fig. 14). Our work therefore reveals the catalytic mechanism of hSMPD2 and advances our understanding of SMases. However, it is not yet possible to determine precisely whether the in vivo substrate of SMPD2 is SM or lyso-PAF, which may require measurements of precise changes in the lipidome in *SMPD2*-knockout mice. Our current study, concentrated on hSMPD2 catalytic properties of sphingomyelin and lyso-PAF at the molecular level, laying the foundation for more in-depth further investigations.

## Methods

### Materials and cell culture

HEK293F (Cat# R79007, Thermo Fisher) cells were cultured in SMM 293-TII medium (Sino Biological Inc; Cat# SMM 293-TII) supplemented with 1 × penicillin/streptomycin (Hyclone, Cat# SV30010) at 37 °C with 5% $CO_2$. DNA for the *Homo sapiens* SMPD2 were cloned from human cDNA and cloned into the pcDNA3.1(-) vector for subsequent protein expression and purification (Forward Primer: GCTAGCGCCACCATG AAGCCCAACTTCTCCCT; Reverse Primer: GTGGTGGAATTCTCAC TTCTCAAACTGGGGGTGGCTCCAT). A twin-Strep affinity tag was inserted after the C-terminus of hSMPD2. Strep-Tactin resin (Cat# 2-1208-500) and D-desthiobiotin (Cat# 2-1000-005) were ordered from IBA. We obtained the Polyethylenimine (PEI, Linear, MW 25000; Cat# 23966) from Polysciences.

### Immunofluorescence co-localization

SY5Y cells (Cat# CRL-2266, ATCC) were cultured in DMEM medium supplemented with 1 × penicillin/streptomycin (Hyclone, Cat# SV30010) and 10% FBS at 37 °C with 5 % $CO_2$. hSMPD2-Flag transient transfection was performed by Lipofectamine™ 3000 reagent (Cat# L3000001, Thermo Fisher) in Opti-MEM™ medium (Cat# 11058021, Thermo Fisher). BODIPY FL labeled sphingomyelin (Cat# D7711) was purchased from Thermo Fisher Inc and Flag antibody (Cat# F3040, dilution: 1:500) was purchased from Merck Inc. Sec61B antibody (Cat# 14648 S, dilution: 1:300) was purchased from Cell Signaling Technology. Giantin antibody (Cat# ab80864, dilution: 1:300) was purchased from Abcam Inc. WGA (Cat# W32466, dilution: 1:300) was purchased from Invitrogen Inc. The fluorescence labeled sphingomyelin was prepared with BSA at a final concentration of 1 mM SM plus 1 mM BSA and stored at −20 °C. After 36–48 h transient transfection, SY5Y cells

was incubated with the SM-BSA complex at a final concentration of 1 μM at 37 °C for 1 h.

The cells were incubated with 4% paraformaldehyde for 15 min at room temperature. The cells were permeabilized with 0.1% Triton X-100 diluted in PBS at room temperature for 3 min followed by incubating with 10% FBS diluted with PBS for 1 h and primary antibody incubation for 1 h. Cells were washed three times with PBS, followed by secondary antibody incubation for 1 h at room temperature. Fluorescence images were acquired using the Olympus FV3000 confocal microscope with single section scanning.

### Protein expression and purification

Transient transfection was performed to heterogeneously express the target protein. In brief, for 1 L culture of HEK293F cells, 1 mg plasmid was pre-incubated with 3 mg 25-kDa linear polyethylenimines (PEIs) (Polysciences) in 50 mL fresh medium for 30 min prior to adding the mixture to cells. The transfected cells were cultured for 48 h before harvest.

For hSMPD2 protein purification, two liters of transfected HEK293F cells were harvested by centrifugation at 3000 g. Cell pellets were resuspended in lysis buffer containing 25 mM Hepes, pH 7.4, and 150 mM NaCl, 1 μg/mL leupeptin, 1.5 μg/mL pepstatin, 0.84 μg/mL aprotinin, 0.3 mM PMSF and lysed by sonication for 5 min. The cell membrane was pelleted after a 100,000 g ultracentrifugation for 1 hour. The membrane was resuspended in buffer containing 25 mM Hepes, pH 7.4, 150 mM NaCl, 1% (w/v) LMNG for 2 h with gentle rotation at 4 °C. After ultra-centrifugation at 100,000 g for 20 min, the supernatant was incubated with Strep-Tactin Sepharose (IBA) for 1 hour with gentle rotation at 4 °C. The resin was washed extensively with wash buffer containing 25 mM Hepes, pH 7.4, 150 mM NaCl, 2 mM DTT, and 0.005% (w/v) LMNG. The target hSMPD2 proteins were eluted with wash buffer plus 5 mM D-Desthiobiotin (IBA) and concentrated to a final volume of approximately 100 μl. The final proteins were applied to size-exclusion chromatography (Superpose-6 10/300 GL, GE Healthcare) in buffer containing 25 mM Hepes, 2 mM $Mg^{2+}$, pH 7.4, 150 mM NaCl and 0.005% LMNG. The peaks corresponding to hSMPD2 proteins were collected for further cryo-microscopy analysis (Supplementary Fig. 2).

The expression and purification procedure of SMPD3 was consistent with that of SMPD2. Briefly, we constructed plasmid of hSMPD3 with C-terminal Strep tag and transiently transfected and overexpressed them in HEK293F cells; subsequently, we purified the corresponding hSMPD3 protein by affinity chromatography and gel filtration chromatography.

### Cryo-electron microscopy

The cryo-EM grids were prepared using a Vitrobot Mark IV (FEI) operated at 8 °C and 100% humidity. For samples of apo hSMPD2, 4 μL aliquots of fresh samples at concentrations of approximately 10 mg/mL were applied onto glow-discharged holey carbon grids, 300 mesh gold (Quantifoil R1.2/1.3). After a waiting time of 5 s, the grids were blotted for 3 s and plunged into liquid ethane for quick freezing.

The cryo-EM grids were screened on a Tecnai Arctica microscope (FEI) operated at 200 kV using a Falcon III 4k × 4k camera (FEI). Qualified grids were transferred to a Titan Krios microscope (FEI) operated at 300 kV for data acquisition and equipped with Gatan K2 Summit detector and GIF Quantum energy filter. Images were automatically recorded using AutoEMation with a slit width of 20 eV for the energy filter and in super-resolution mode at a nominal magnification of 105,000×, corresponding to a calibrated pixel size of 0.8374 Å at object scale, and with defocus ranging from −1.4 to −1.9 μm. Each stack was exposed for a total of 1.28 s with an exposure of 0.04 s for each of 32 frames; Total dose rate for each stack was about 50 e⁻/Å².

## Image processing

Simplified flowcharts for data processing of hSMPD2 were summarized in Supplementary Fig. 4. In total, 1953 movie stacks were collected for hSMPD2 samples. Motion correction was performed using MOTIONCORR2[41], generating summed micrographs with or without dose weighting. CTFFIND4[42] was used to estimate the contrast transfer function (CTF) parameters and produce the CTF power spectrum on the basis of summed micrographs from MOTIONCORR2. Particles were auto-picked on summed micrographs from MOTIONCORR2 using RELION-3[43].

1334 k particles were auto-picked from 1953 micrographs and two rounds of 2D classifications were performed to exclude noise and other bad particles. 836 k particles from qualified 2D averages were selected for further 3D analysis. Ahead of 3D classification, a round of refinement was applied on the whole particle sets using RELION-3. Three rounds of 3D classification with C1 symmetry generated 220 k particles with good signal. Each particle was re-centered using the in-plane translations measured in 3D refinement and re-extracted from the motion-corrected integrated micrographs. Gctf was used to refine the local defocus parameters[44]. The final particles were processed by auto-refine with soft mask and C2 symmetry imposed using RELION-3, resulting in a 3.07-Å resolution map of hSMPD2 with C2 symmetry (Supplementary Fig. 4).

## Model building

Before model building, model of full-length hSMPD2 was predicted from the alpha fold server. Sequence alignment and secondary structure prediction of hSMPD2 were used to aid the model building. The predicted model of hSMPD2 was docked into the cryo-EM map with a resolution of 3.07 Å in Chimera and manually adjusted in Coot to acquire the atomic model of hSMPD2[45,46]. Model refinement was performed on the main chain of the two atomic models using the real_space_refine module of PHENIX[47] with secondary structure and geometry restraints to avoid over-fitting. All reported resolutions are based on the gold-standard FSC = 0.143 criteria[48], and the final FSC curves are corrected for the effect of a soft mask using high-resolution noise substitution[49]. Final density maps were sharpened using RELION, and local resolution maps were calculated using ResMap[50]. Models with ligands and phospholipids were subjected to global refinement and minimization in real space refinement using PHENIX.

## Amplex red sphingomyelinase assay

hSMPD2 activity was quantified with the Amplex Red Sphingomyelinase Assay Kit (Cat# A12220; Thermal Scientific) at 37 °C, in which the 10-acetyl-3,7-dihydroxyphenoxazine was utilized to monitor sphingomyelinase activity. 5 mM Triton X-100 solubilized sphingomyelin was utilized as the substrate. Purified proteins were solubilized in buffer which contains 25 mM Hepes pH = 7.4, 150 mM NaCl, 0.005% LMNG, 2 mM $Mg^{2+}$. The reaction was started by adding 100 μL of the Amplex Red reaction mixture (100 mM Tris pH 7.4, 10 mM $MgCl_2$, 2 U/mL horseradish peroxidase, 0.2 U/mL choline oxidase, 8 U/mL alkaline phosphatase, 0.5 mM sphingomyelin, 0.2% (w/v) Triton X-100) to 100 μL protein solution which contains 125 nM dimeric hSMPD2. After rotated shaking for 15 s, the emission signal at 590 nm was measured by a Perkin/Elmer plate reader with excitation at 530 nm. Obtained data was analyzed by GraphPad Prism 8 (GraphPad Software, Inc.).

## In vitro pull-down assay

To facilitate the verification of the dimer interface residues, we constructed Q91A, E92A, Y105A, W112A, F113A, L356A, W367A, and T359A mutants of hSMPD2 with Strep tag and Flag tag. In brief, we transfect 50-mL HEK293F cells with the following combinations: WT hSMDP2-Strep and WT hSMPD2-Flag, point mutation hSMPD2-Strep and point mutation hSMPD2-Flag. Transfected cells were lysed in 1-mL lysis buffer [25 mM tris (pH 7.8), 150 mM NaCl, 1 mM EGTA, cOmplete

protease inhibitors] with 1% digitonin. The cell lysate was incubated for 30 min on ice and centrifuged for 10 min at 4 °C at 20,000 g. The supernatant was incubated with anti-Strep magnetic agarose (Thermo Fisher Scientific) for 2 h at 4 °C. The beads were collected on the magnet, washed three times with 1 mL of lysis buffer containing 0.1% digitonin, and eluted with 150 μl of SDS-gel loading buffer for Western blot. For Western blot analysis, proteins were subjected to a 4 to 20% SDS–polyacrylamide gel electrophoresis gel (GenScript) and transferred onto a polyvinylidene difluoride membrane (Millipore). Membranes were detected with the indicated antibodies.

## Molecular docking of sphingomyelin along the hydrophobic groove of hSMPD2

The initial conformation of sphingomyelin was manually placed along the hydrophobic groove of hSMPD2 and the RosettaLigand docking protocol[51] was used for molecular docking with a library of 300 conformations of sphingomyelin selected from CHARMM-GUI archive for biomembrane systems[52].

The script for the docking protocol is available at: https://github.com/yuebinzhang/Yi_et_at_hSMPD2_Nat_Comm/tree/main/RosettaLigand_docking/.

The conformation with a minimum distance between the magnesium ion and the non-bridging oxygen atom of the phosphate group of sphingomyelin was used for further hydrolysis mechanism investigation.

## Explicit membrane MD simulations of dimeric hSMPD2 binding with the docked sphingomyelin

The dimeric architecture of hSMPD2 binding with the docked sphingomyelin was used as the initial conformation to construct the molecular dynamics simulations system in the explicit membrane bilayer environment using CHARMM-GUI. The composition of the bilayer (number of lipids) was listed in Supplementary Table 2.

The CHARMM36 force field[53] was employed for both protein and lipids and the TIP3P water model was used to solvate the system and 0.1 M NaCl ions were used to balance the net charges of the system, which yields the initial simulation box of 116.88 Å × 116.88 Å × 175.0 Å with a total of atom number of 225,288. The system was equilibrated with the standard six-steps protocol[54,55] from CHARMM-GUI simulation input files using NAMD 3.0[56].

## Hydrolysis mechanism investigation with QM/MM free energy calculations

For QM/MM simulations, we used the CHAMBER command to covert the parameter files as well as psf and pdb files into AMBER format and the AMBER package[57] and the QUantum Interaction Computational Kernel (QUICK) program[58] which takes the advantage of GPU-acceleration in electronic structure calculations was used for QM/MM simulations. The QM region contained 106 atoms including link-hydrogen atoms and the side chains (without Cβ) of N15, E48, E49, D111, K116, D178, D238 and H272 and the phosphocholine group of sphingomyelin and $Mg^{2+}$ ion as well as two water molecules. The PBE0 functional[59] and def2-SVP basis sets were used for QM calculations and the PLUMED free energy calculation library[60] was employed for enhanced sampling QM/MM simulations.

We first performed QM/MM steered MD (SMD) simulations to drive to catalytic reaction in the forward direction from the reactant state to the product state by defining the reaction coordinate as the distance difference (dd) between the forming bond and the scissile bond. The QM/MM SMD simulations were performed for 2.5 ps using 1 fs timestep with the boundary of dd from −2 Å to 2.4 Å with a force constraint of 100 kcal/mol·Å². Then, umbrella sampling (US) was employed to estimate the free energy profile along the reaction coordinate, by which 45 bins from −2 Å to 2.4 Å with 0.1 Å increment were used. The initial conformation of each bin was extracted from

QM/MM SMD simulation trajectory and 10 ps QM/MM US simulations were conducted for each bin and the last 3 ps trajectories were used for analysis. For using lyso-PAF as a substrate, we replaced the sphingo-myelin by lyso-PAF for the initial conformation of each bin while keeping the rest of the setting parameters as the same as that of sphingomyelin for parallel comparison. The weight of each config-uration was calculated according to PLUMED introduced by belfast-4, and the histogram command was used for reweighing. The script for reweighting was available at https://github.com/yuebinzhang/Yi_et_at_hSMPD2_Nat_Comm/tree/main/wham_plumed/plumed_fes.dat.

## Statistics & reproducibility

Statistics analysis was performed using the GraphPad Prism 8 software. Data are represented as individual values, mean, mean +/−SEM as described in figure legends. Group sizes ($n$) are indicated in related figure legends. No data were excluded except cryo-EM data, in which bad micrographs or particles were excluded to reach a high resolution. The pull-down assay is reproducible and was repeated more than three times independently.

## Reporting summary

Further information on research design is available in the Nature Portfolio Reporting Summary linked to this article.

## Data availability

The 3D cryo-electron microscopy density map has been deposited in the Electron Microscopy Data Bank (EMDB), with accession code 35948. The coordinates of atomic models have been deposited in the Protein Data Bank (PDB), with the accession code 8J2F. Structure used for comparative analysis in this manuscript can be found with follow-ing PDB accession code: 5UVG for human SMPD3; 5FIC for human acid SMase; 1ZWX for Smcl from *Listeria ivanovii*; 2DDT for sphingomyelin phosphodiesterase from *Bacillus cereus*; 3I48 for beta toxin from *Sta-phylococcus aureus*. All protein sequences used in this study are available at Uniprot (https://www.uniprot.org/) with the following accession codes: O60906 for human SMPD2; Q9NY59 for human SMPD3. D6MZJ6 for MA-nSMase from *Mus musculus*; P40015 for ISC1 from *S. cerevisiae*; Q9RLV9 for nSMase from *Listeria ivanovii*; P09599 for nSMase from *B. cereus*; P09978 for nSMase from *Staphylococcus aureus*. Source data are provided as a Source Data file.

## Code availability

Codes used for docking, reweighting, and MD simulation system pre-paration are available at the Github repository: https://github.com/yuebinzhang/Yi_et_at_hSMPD2_Nat_Comm. Any additional MD simula-tion data required for analysis is available upon request.

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

## Acknowledgements

We thank the Tsinghua University Branch of China National Center for Protein Sciences (Beijing) for providing the cryo-EM facility support. The computation was completed on the Yanglab GPU workstation. This work was supported by funds for M.Y. from the National Key R&D Program of China (2022YFA1302701), the National Natural Science Foundation of China (32030056), the Tsinghua-Foshan Innovation Special Fund (TFISF-2022THFS6122), the King Abdullah University of Science and Technology (KAUST) Office of Sponsored Research (OSR) under Award (OSR-2020-CRG9-4352), and by funds for Y.Z. from the Strategic Priority Research Program of Chinese Academy of Sciences (Grant No. XDB 37000000).

## Author contributions

M.Y. directed the study. J.Yi, S.Z., B.Q. did the protein purification, detergent screening, performed EM sample preparation, data collection and structural determination; Y.Z. and G.L. performed simulations and analysis; J.Yi performed the Amplex Red Sphingomyelinase Assay with the help of R.L, J.Yin, X.C. and J.H.; M.Y., S.Z., J.Yi and Y.Z. built the model, drew the figures, and wrote the manuscript. All authors contributed to the discussion of the data and editing the manuscript.

## Competing interests

The authors declare no competing interests.
