## [Peer Review File · Nature Communications]

REVIEWER COMMENTS

Reviewer #1 (Remarks to the Author):

Yi et al present the cryo-em structure of the full-length neutral sphingomyelinase nSMase1 (product of the SMPD2 gene) in a detergent-solubilized form. The dimer creates a very interesting pocket for lipid binding in the dimer interface. The authors carry out numerous mutations to probe the function of the enzyme. They carry out MM simulations and QM calculations to probe the mechanisms of substrate binding and the mechanism of hydrolysis. Overall, the included experimental work is interesting, novel, important, and well done. The interpretation and presentation of the data, however, detracts from the quality of this manuscript.

Major comments:

1. A Figure for the proposed reaction mechanism is not presented. The QM/MM calculations seem to relate to this, but no clear description or figure of the mechanism is provided. Figure 7 is a schematic and does not show the mechanism described in the MS (including lines 300-307). Previous papers on this family of enzymes have proposed mechanisms. These should be included and described/compared. If this cannot be done, remove "catalytic mechanism" from the title and text.

2. Line 128-131. Mutants L356A, T359A and W367A (and others) do not affect activity (Fig 2d). The assumption is that these mutations disrupt the dimer (line 147-148) but this is not shown. The pocket for lipid binding presumably depends on the dimer. So the more likely explanation is that the "dimerization" mutants do not disrupt the dimer. Can an assay for dimerization be developed? Instead of saying that T359 and W367 are important for stabilizing the dimer, simply state that they are in contact. Importantly, no data is included to show that the dimer is functionally important and required for activity. Overall, the section on dimerization and the mutants used to probe this is confusing and should be re-written.

3. The section "Structure divergence among nSMases" (Lines 271-292, Figures 6, S4) has many problems, including: 1) nSMase is a different fold family (not related by evolution) with a different active site/catalytic mechanism - the open/closed comparison is like comparing apple and oranges; 2) hSMPD3 has an Nterminal helical anchor, but this is not a saposin domain (line 278); 3) Line 281 is wrong: the DK loop is conserved even in the bacteria(see Fig, 3c of ref 28) – the alignment presented in Fig S4 has many errors. A better alignment can be generated with hhpred, but even better would be a structure-based sequence alignment of the relevant structures.

A more direct discussion/comparison of nSMase1 and nSMase2 should be included, with a more careful consideration of the results from previous papers.

4. Discussion: Only the first paragraph of the Discussion relates to the work presented here. The rest is about the biological relevance of SMPD2, with no insights provided by the data in this MS.

Minor comments:

5. Please show the density for the four "lipids" in the cleft (line 136) as a supplementary figure. It is more likely that these are LMNG detergent molecules.

6. Line 53: "recent studies" refers to four papers; these were published in 1998, 2000, 2006 and 2010. The "recent" should not be used here. Simply "There are four human nSMases ..."

7. Lines 89-91: The suggestion of phospholipase activity for SMPD2 was published in 1999 (ref 26), but this protein has long since been recognized as a sphingomyelinase making the first guess at activity irrelevant. The setup for the rationale "Thus, to elucidate ..." does not reflect the actual justification for the study.

8. Line 155: how do you know the ion is a magnesium?

9. Fig. 4A is unclear and very difficult to interpret.

10. The included PDB validation report was generated with the standalone wwPDB validation server and is not acceptable for manuscript review.

11. Methods: please include the expression of hSMPD3 (used in Fig 1b).

12. SI Fig. 4. Caption: what do the two types of boxed residues indicate?

13. Suppl. Fig 6b: I think N5 should be N15.

Reviewer #2 (Remarks to the Author):

In this study, Yi et al describe the structure of SMPD2 from human using cryo-EM and Interestingly, the structure reveals trans-membrane helices, which is different than what is postulated for SMPD3. This is a well conducted and described study and makes an important contribution to the literature.

Major points.

1. A major issue concerning the metabolic function of SMPD2 is the paucity of results showing that it functions as a sphingomyelinase in cells. (for example, reference 18 does not demonstrate a role for SMPD2 in directly regulating ceramide levels; reference 19 does not study SMPD2 at all). This does not detract from the structural analysis, but for biochemical action, it would seem necessary to study a better documented endogenous substrate (lysoPAF). Thus, conducting simulation and biochemical studies with lyso-PAF is critical.

a. Note that the in vitro ability of SMPD2 to hydrolyze SM has never been under question, so there is little need to 'verify' that.

2. The molecular simulation (Fig 3c) does not appear to show transmembrane spanning. This is an important biochemical point that should be addressed experimentally (e.g. sensitivity to proteases on side vs the other of the membrane).

Specific points

1. Do residues Q91, E92, L356, W267, and T359 affect dimerization?

2. The loss of activity with Y105A, W112A, and F113A could be due to loss of protein stability, etc, so the conclusion that they are required for hydrolysis (p6) is premature.

3. Page 9 middle (should be Airola et al not Michael et al)

Reviewer #3 (Remarks to the Author):

The manuscript by Yi et al reports the first experimental structure for full length hSMPD2. The structure is well refined and the supplemental information shows the density maps allowed unambiguous refinement of the model. A novel dimeric architecture is observed with the transmembrane segments forming extensive contacts to mediate dimer formation. A series of point mutations are carried out to identify roles of residues in catalysis. Some of these match results from previous experiments with the yeast homolog Isc1, some reveal novel roles for residues in dimer formation, and some have no effect (e.g. residues in the transmembrane helix dimer). A docking analysis and molecular dynamics are used to simulate the catalytic process. Point mutations of the magnesium coordinating residues eliminate activity as anticipated. Mutations in a previously identified putative substrate recognition motif (the D-K loop) eliminate or nearly eliminate catalytic activity. However, a mechanistic role for these residues in hydrolysis is not mentioned.

The manuscript advances the understanding of the SMPD class of enzymes by revealing a dimeric architecture. However, the insights are somewhat limited in that the role of SMPD2 as a sphingomyelinase in cells is not clearly established, with evidence that SMPD2 can convert sphingomyelin to ceramide *in vitro*, but does not convert sphingomyelin to ceramide in cells. Additional insight could be obtained from a more detailed comparison between hSMPD2 and the well characterized *S. cerevisiae* Isc1p, which both share the same domain architecture (e.g. an N-terminal catalytic domain and two C-terminal transmembrane helices). Additionally, more details about catalysis beyond the role of magnesium binding residues or water coordinating residues would enhance the mechanistic findings, and more detailed biochemical experiments/analysis of residues proposed in sphingomyelin binding.

Major points

1. The authors cite a previous study that found nSMase1 hydrolyzes the lyso-phospholipid lyso-PAF. This study found that nSMase1 can hydrolyze SM *in vitro*, but that in cells nSMase1 does not affect sphingomyelin levels since its subcellular localization is at the ER, which does not overlap with the localization of sphingomyelin. Thus, although nSMase1 can hydrolyze SM *in vitro* and has comparable activity to nSMase2 (as demonstrated by the authors), can it really be classified as a sphingomyelinase?
2. The authors demonstrate experimentally that residues in the D-K loop are important for catalysis but the docking sections and MD simulations do not mention a role for these residues and how they participate in either sphingomyelin binding or catalysis. Some discussion appears relevant here.
3. Since SMPD2 shares the same domain architecture as Isc1p (e.g. an N-terminal catalytic domain and two C-terminal transmembrane helices) it would be useful to include more discussion about whether key residues are conserved in Isc1p and/or the implications to Isc1p function. As currently written, the manuscript focuses on comparing the SMPD2 structure to Isc1p, SMPD3 and SMPD4. However, SMPD3 only shares a similar catalytic domain and SMPD4 belongs to a different protein fold. Thus, comparisons of the catalytic domains of SMPD2 and SMPD3 are interesting, but any comparison between SMPD2 and SMPD4 do not seem very relevant.
4. All the point mutations appear to be characterized at single concentration of sphingomyelin, which limits the mechanistic insight of these residues in catalysis. To better understand the role of these residues (sphingomyelin affinity vs. catalysis rate) it would be useful (if the authors' assay is compatible) to determine hSMPD2 activity at different molar ratios of sphingomyelin to Triton X-100 to see if the point mutations

affect apparent K_M or k_{cat} .

5. Lines 277-279: The statement "whereas hSMPD3 and ASMases have been proposed to adopt a similar dimeric structure through the N-terminal juxtamembrane saposin domains" should be reworded. ASMase has a saposin domain, hSMPD3 has a juxtamembrane region. I'm not aware of evidence for dimer formation in hSMPD3. Is there a reference for this?

6. Lines 280-281: The statement "The second deviation is that the D-K loop domain identified in hSMPD2 is not conserved among hSMPD families" should be reworded and/or put into context. The DK-loop is conserved in SMPD3, SMPD5, bacterial nSMases, and yeast Isc1p; it is only not conserved in SMPD4, which belongs to a different enzyme family. The sequence alignment included shows an alignment between SMPD2,3,4 which doesn't provide much information since SMPD4 has a different protein fold and sequence homology is not relevant.

7. Line 153: "In addition to nSMase3" should be "With the exception of nSMase3". Also consider adding a citation for this if relevant.

8. Lines 144-148: The authors state: "hSMPD2 with Q91A, E92A, L356A, W367A, or T359A point mutations each had similar sphingomyelin hydrolysis activity compared to wild type hSMPD2 (Fig. 2d and Supplementary Fig. 5a), indicating that these residues predominantly participate in protein dimerization rather than in substrate hydrolysis". I'm assuming that mutation of these residues does not eliminate dimer formation? Are the authors implying that dimer formation is not important for hSMPD2 activity?

9. Lines 157-159: "Residue N15 is conserved in all nSMases and in their homolog in *Saccharomyces cerevisiae*, Isc1p, whereas residue E49 is only conserved among hSMPD2, hSMPD4, and Isc1p". Are the authors sure that this glutamate residue is not conserved in hSMPD3? I would suspect it is not conserved in SMPD4 because that is a different protein fold, but it would be very surprising if SMPD3 does not conserve a key residue for Mg coordination.

10. Line 181: consider rewording "the winning pose". This phrasing does not make clear why this pose was chosen. There must be a logical reason why it was selected and by explicitly stating the reason, it would make clear why it was the binding pose most consistent with catalysis.

11. The I19A mutation appears to retain ~20% activity, not 5% (as shown in Fig. 3e). Is this a typo?

12. Lines 287-288: The statement "but such a conformation change in the saposin domain was not observed in the current hSMPD2 model" implies that SMPD2 has a saposin domain, which it does not. SMPD1 (acid sphingomyelinase) has a saposin domain but not SMPD2.

REVIEWER COMMENTS

Reviewer #1 (Remarks to the Author):

Yi et al present the cryo-em structure of the full-length neutral sphingomyelinase nSMase1 (product of the SMPD2 gene) in a detergent-solubilized form. The dimer creates a very interesting pocket for lipid binding in the dimer interface. The authors carry out numerous mutations to probe the function of the enzyme. They carry out MM simulations and QM calculations to probe the mechanisms of substrate binding and the mechanism of hydrolysis. Overall, the included experimental work is interesting, novel, important, and well done. The interpretation and presentation of the data, however, detracts from the quality of this manuscript.

Major comments:

1. A Figure for the proposed reaction mechanism is not presented. The QM/MM calculations seem to relate to this, but no clear description or figure of the mechanism is provided. Figure 7 is a schematic and does not show the mechanism described in the MS (including lines 300-307). Previous papers on this family of enzymes have proposed mechanisms. These should be included and described/compared. If this cannot be done, remove “catalytic mechanism” from the title and text.

Response: We appreciate the reviewer’s comment for helping us improve the quality of our work. In the revised manuscript, we replaced the origin figure 4 with a schematic representation of the hypothetical pathway for the hydrolysis reaction.

The catalytic mechanisms of acid sphingomyelinase with two zinc ions observed at the catalytic center were proposed by previous published papers (doi:10.1038/ncomms13082; DOI: 10.1038/ncomms12196). However, the functions of neutral sphingomyelinases are Mg^{2+} dependent and previously reported homologous structures of human neutral sphingomyelinases, including the *S. aureus* sphingomyelinase (PDB code 3I48) and *B. cereus* sphingomyelinase (PDB code 2DDT), shown a single Mg^{2+} at the catalytic center. In our current hSMPD2 structure, we observed a magnesium ion located in a negatively charged cleft in the CAD that is further coordinated by residues N15 and E49 in the active site. The catalytic mechanisms between two-metal-mediated phosphodiester bond cleavage and single-metal-mediated one are quite different, thus, could not be comparable.

As far as we searched the literatures, the catalytic mechanisms of neutral sphingomyelinases have not been elucidated before both experimentally or theoretically. We also tried to solve the structures of hSMPD2 in different states by adding ceramide or sphingomyelin before cryo-sample preparation. However, we were failed to obtain these states (the ceramide-bound and the sphingomyelin-bound structures). Thus, in this study, a combination of different computational methods including molecular docking, classical MD simulations, QM/MM free energy simulations have been used to hypothesises the phosphodiester bond cleavage mechanism catalyzed by single Mg(II) center of neutral sphingomyelinases. We believe our theoretical work would provide new insight in understanding the catalytic mechanism and direct further experimental validation for the proposed phosphodiester hydrolysis mechanisms of the neutral sphingomyelinase family.

2. Line 128-131. Mutants L356A, T359A and W367A (and others) do not affect activity (Fig 2d). The assumption is that these mutations disrupt the dimer (line 147-148) but this is not shown. The pocket for lipid binding presumably depends on the dimer. So the more likely explanation is that the “dimerization” mutants do not disrupt the dimer. Can an assay for dimerization be developed? Instead of saying that T359 and W367 are important for stabilizing the dimer, simply state that they are in contact. Importantly, no data is included to show that the dimer is functionally important and required for activity. Overall, the section on dimerization and the mutants used to probe this is confusing and should be re-written.

Response: We thank for the comments and we have corrected the inappropriate description in the revised manuscript. Seeking experimental validation for dimerization of the residues, in vitro pull-down assays were performed with extracts from cells co-expressing Strep-tagged hSMPD2 and Flag-tagged hSMPD2 by using Strep-tagged proteins as “bait” protein.

The expression level of Q91A, L356A, T359A, and W367A are comparable to WT protein, whereas E92A mutants largely reduced the protein expression level compared with the wildtype, resulting in a decreased dimer stability. Multiple unbiased MD simulations (5 x 50 ns) after the

equilibration phase indicates that E92 and K168 could form a stable salt bridge throughout the MD simulations, further verifying the essential role of this salt bridge interaction in dimer formation

According to the reviewer’s suggestions and our revised experiments, E92-K168 salt bridge interaction is important in stabilizing the dimeric architecture, whereas the other residues are simply in contact.

3. The section “Structure divergence among nSMases” (Lines 271-292, Figures 6, S4) has many problems, including: 1) ASMase is a different fold family (not related by evolution) with a different active site/catalytic mechanism - the open/closed comparison is like comparing apple and oranges; 2) hSMPD3 has an Nterminal helical anchor, but this is not a saposin domain (line 278); 3) Line 281 is wrong: the DK loop is conserved even in the bacteria(see Fig, 3c of ref 28) – the alignment presented in Fig S4 has many errors. A better alignment can be generated with hhpred, but even better would be a structure-based sequence alignment of the relevant structures. A more direct discussion/comparison of nSMase1 and nSMase2 should be included, with a more careful consideration of the results from previous papers.

Response: We thank for the concerning about this issue. According to the helpful suggestions, we have deleted the comparisons between hSMPD2 and ASMases, and mainly focused on the comparisons with SMPD3 and Isc1p. We have now corrected the sequence alignment from the HHpred web-server by combining the secondary structure prediction and the reported nSMases structures. Moreover, we have corrected the descriptions regarding the N-terminal helical anchor in hSMPD3 in the revised manuscript.

4. Discussion: Only the first paragraph of the Discussion relates to the work presented here. The rest is about the biological relevance of SMPD2, with no insights provided by the data in this MS.

Response: We thank for the comments about discussion section. We have removed the irrelevant descriptions and focused mainly on the description of the catalytic mechanism and new insights from our current structural and simulation studies.

Minor comments:

5. Please show the density for the four “lipids” in the cleft (line 136) as a supplementary figure. It is more likely that these are LMNG detergent molecules.

Response: We thank for the concerning about the lipid’s densities in the manuscript. According to the helpful suggestions, we have shown the lipid density below as a supplementary figure. Also, we have examined the possibility as LMNG molecules, however, which is too big to accommodate the cryo-EM densities and might trigger spatial clashes to the surrounding residues.

6. Line 53: “recent studies” refers to four papers; these were published in 1998, 2000, 2006 and 2010. The “recent” should not be used here. Simply “There are four human nSMases ...”

Response: We thank for the comments and we have corrected the inappropriate description in the revised manuscript.

7. Lines 89-91: The suggestion of phospholipase activity for SMPD2 was published in 1999 (ref 26), but this protein has long since been recognized as a sphingomyelinase making the first guess at activity irrelevant. The setup for the rationale “Thus, to elucidate ...” does not reflect the actual justification for the study.

Response: We thank you for your helpful comments and we have corrected the inappropriate descriptions in the revised manuscript. First, we have added two recent studies on the role of SMPD2 in ceramide production and sphingolipid hydrolysis to the Introduction. Second, we show fluorescence co-localization experiments of SM and SMPD2 in cells. In addition, we investigated the kinetic parameters of SMPD2 on lyso-PAF and SM.

8. Line 155: how do you know the ion is a magnesium?

Response: We thank for the concerning about this issue. We have shown the density for magnesium and surrounding residues as supplementary figures. All neutral sphingomyelinases possess a Mg^{2+}

binding domain and their functions are Mg^{2+} dependent (PMID: 21035485; PMID: 20552297; PMID: 10823942). The reported homologous structures, including the *S. aureus* sphingomyelinase (PDB code 3I48) and *B. cereus* sphingomyelinase (PDB code 2DDT), contains Mg^{2+} in the catalytic center. Besides, we have added 2 mM Mg^{2+} during the protein purification procedure.

9. Fig. 4A is unclear and very difficult to interpret.

Response: We thank the reviewer for the valuable comments. Accordingly, we have revised and uploaded the new figure to better interpret our catalytic mechanism.

10. The included PDB validation report was generated with the standalone wwPDB validation server and is not acceptable for manuscript review.

Response: We thank the reviewer for the valuable comments. Accordingly, we have uploaded the PDB file to the validation server and the status of the entry now is HPUB (Hold for publication). The validation report was also uploaded as a supplementary file.

11. Methods: please include the expression of hSMPD3 (used in Fig 1b).

Response: We thank the reviewer for the suggestions. We have included the expression method of hSMPD3 in the methods section.

12. SI Fig. 4. Caption: what do the two types of boxed residues indicate?

Response: We thank for the concerning about this issue. We have now corrected the sequence alignment from the HHpred web-server by combining the secondary structure prediction and the reported nSMases structures.

13. Suppl. Fig 6b: I think N5 should be N15.

Response: We thank for the comments and we have corrected the mistake in the revised manuscript.

Reviewer #2 (Remarks to the Author):

In this study, Yi et al describe the structure of SMPD2 from human using cryo-EM and Interestingly, the structure reveals trans-membrane helices, which is different than what is postulated for SMPD3. This is a well conducted and described study and makes an important contribution to the literature.

Major points.

1. A major issue concerning the metabolic function of SMPD2 is the paucity of results showing that it functions as a sphingomyelinase in cells. (for example, reference 18 does not demonstrate a role for SMPD2 in directly regulating ceramide levels; reference 19 does not study SMPD2 at all). This does not detract from the structural analysis, but for biochemical action, it would seem necessary to study a better documented endogenous substrate (lysoPAF). Thus, conducting simulation and biochemical studies with lyso-PAF is critical.

a. Note that the in vitro ability of SMPD2 to hydrolyze SM has never been under question, so there is little need to ‘verify’ that.

Response: We thank for the concerning and comments about the metabolic function of SMPD2; First, we have added two recent studies on the role of SMPD2 in ceramide production and sphingomyelin hydrolysis to the Introduction section. Xing. et al. revealed that A β treatment increased the expression and activity of nSMase1, which results in decreased sphingomyelin level and upregulated ceramide level (PMID: 36453415). Dolma. et al. observed reduced ceramide level specifically at the ER in SMPD2 knockdown cell, which further affects cellular fitness under ER stress. Second, we show fluorescence co-localization experiments of SM and SMPD2 in cells. Fluorescence co-localization between SM and overexpressed SMPD2 were observed on the ER and plasma membrane. In addition, we have also investigated the kinetic parameters of in vitro purified SMPD2 on lyso-PAF and SM. The Km value for SM was comparable to that of lyso-PAF, in agreement with the previous report

2. The molecular simulation (Fig 3c) does not appear to show transmembrane spanning. This is an important biochemical point that should be addressed experimentally (e.g. sensitivity to proteases on side vs the other of the membrane).

Response: In the revised manuscript, a combination of different computational methods including molecular docking, classical MD simulations, QM/MM free energy simulations have been used to hypothesize the phosphodiester bond cleavage mechanism catalyzed by single Mg(II) center of neutral sphingomyelinases. We have conducted MD simulations using the full-length dimeric architecture of hSMPD2 binding with the docked sphingomyelin in the explicit membrane environment composed of a SM/POPS/POPC bilayer. The new MD simulation results further reveal the essential role of D111-K116 loop in stabilizing and facilitating SM hydrolysis reactions. In addition, we also found an important salt bridge formed by E92 and K168 of the other protomer in maintaining the dimeric architecture stability of hSMPD2.

Specific points

1. Do residues Q91, E92, L356, W267, and T359 affect dimerization?

Response: We thank for the comments about these residues. Seeking experimental validation for dimerization of the residues, in vitro pull-down assays were performed with extracts from cells co-expressing Strep-tagged hSMPD2 and Flag-tagged hSMPD2 by using Strep-tagged proteins as “bait” protein.

The expression level of Q91A, L356A, T359A, and W367A are comparable to WT protein, whereas E92A mutants largely reduced the protein expression level compared with the wildtype, resulting in a decreased dimer stability. Multiple unbiased MD simulations (5 x 50 ns) after the equilibration phase indicates that E92 and K168 could form a stable salt bridge throughout the MD simulations, further verifying the essential role of this salt bridge interaction in dimer formation

According to the reviewer's suggestions and our revised experiments, E92-K168 salt bridge interaction is important in stabilizing the dimeric architecture, whereas the other residues are simply in contact.

2. The loss of activity with Y105A, W112A, and F113A could be due to loss of protein stability, etc, so the conclusion that they are required for hydrolysis (p6) is premature.

Response:

Similarly, we also performed pull-down assay on these amino acids and showed that mutations in these amino acids did not affect protein expression and dimeric forms. These amino acids may contribute to stabilize the hydrophobic groups of sphingomyelin from our structural and simulation studies, and therefore mutation of these amino acids robustly diminished sphingomyelin hydrolysis compared to wild type enzyme.

3. Page 9 middle (should be Airola et al not Michael et al)

Response: We appreciate your comments and we have corrected this error in the revised manuscript.

Reviewer #3 (Remarks to the Author):

The manuscript by Yi et al reports the first experimental structure for full length hSMPD2. The structure is well refined and the supplemental information shows the density maps allowed unambiguous refinement of the model. A novel dimeric architecture is observed with the transmembrane segments forming extensive contacts to mediate dimer formation. A series of point mutations are carried out to identify roles of residues in catalysis. Some of these match results from previous experiments with the yeast homolog *Isc1*, some reveal novel roles for residues in dimer formation, and some have no effect (e.g. residues in the transmembrane helix dimer). A docking analysis and molecular dynamics are used to simulate the catalytic process. Point mutations of the magnesium coordinating residues eliminate activity as anticipated. Mutations in a previously identified putative substrate recognition motif (the D-K loop) eliminate or nearly eliminate catalytic activity. However, a mechanistic role for these residues in hydrolysis is not mentioned.

The manuscript advances the understanding of the SMPD class of enzymes by revealing a dimeric architecture. However, the insights are somewhat limited in that the role of SMPD2 as a sphingomyelinase in cells is not clearly established, with evidence that SMPD2 can convert sphingomyelin to ceramide in vitro, but does not convert sphingomyelin to ceramide in cells. Additional insight could be obtained from a more detailed comparison between hSMPD2 and the well characterized *S. cerevisiae* *Isc1p*, which both share the same domain architecture (e.g. an N-terminal catalytic domain and two C-terminal transmembrane helices). Additionally, more details about catalysis beyond the role of magnesium binding residues or water coordinating residues would enhance the mechanistic findings, and more detailed biochemical experiments/analysis of residues proposed in sphingomyelin binding.

Major points

1. The authors cite a previous study that found nSMase1 hydrolyzes the lyso-phospholipid lyso-PAF. This study found that nSMase1 can hydrolyze SM in vitro, but that in cells nSMase1 does not affect sphingomyelin levels since its subcellular localization is at the ER, which does not overlap with the localization of sphingomyelin. Thus, although nSMase1 can hydrolyze SM in vitro and has comparable activity to nSMase2 (as demonstrated by the authors), can it really be classified as a sphingomyelinase?

Response: We appreciate your comments and concerns about this issue. First, we performed fluorescence colocalization experiment in SY5Y cells by overexpressing hSMDP2. The fluorescence labeled sphingomyelin localized to the plasma membrane and ER, whereas the overexpressed hSMPD2 localized to the ER and partially to the plasma membrane. Fluorescence co-localization between SM and SMPD2 were observed on the ER and plasma membrane. Second, we have added two recent studies on the role of SMPD2 in ceramide production and sphingolipid hydrolysis to the Introduction section. Xing. et al. revealed that A β treatment increased the expression and activity of nSMase1, which results in decreased sphingomyelin level and upregulated ceramide level (PMID: 36453415). Dolma. et al. observed reduced ceramide level specifically at the ER in SMPD2 knockdown cell, which further affects cellular fitness under ER

stress. Third, our attempt to perform the isotope quantitative experiment of SM in cells was hampered due to the inaccessible to purchase the isotopically labeled SM. In addition, we have also investigated the kinetic parameters of in vitro purified SMPD2 on lyso-PAF and SM, which further reveals that the K_m value for SM was comparable to that of lyso-PAF.

2. The authors demonstrate experimentally that residues in the D-K loop are important for catalysis but the docking sections and MD simulations do not mention a role for these residues and how they participate in either sphingomyelin binding or catalysis. Some discussion appears relevant here.

Response: We thank for the comments on the role of D-K loop region. As illustrated in our revised sequence alignment, residues Asp and Lys are conserved from human to bacteria, signifying a pivotal role in catalysis process. The current MD simulations indicate that, during sphingomyelin binding and catalysis, the side chain of lysine tends to bind and stabilize the phosphate group of sphingomyelin. The minimum distance between amine of lysine and the phosphate group of SM is around 2.6 angstrom. Thus, D-K loop region is essential for the SM

hydrolysis process, i.e., the conserved lysine binds and forms interactions with the phosphate group of SM during SM binding, whereas the aromatic residues of the D-K loop region help stabilize the hydrophobic tail of SM.

3. Since SMPD2 shares the same domain architecture as Isc1p (e.g. an N-terminal catalytic domain and two C-terminal transmembrane helices) it would be useful to include more discussion about whether key residues are conserved in Isc1p and/or the implications to Isc1p function. As currently written, the manuscript focuses on comparing the SMPD2 structure to Isc1p, SMPD3 and SMPD4. However, SMPD3 only shares a similar catalytic domain and SMPD4 belongs to a different protein fold. Thus, comparisons of the catalytic domains of SMPD2 and SMPD3 are interesting, but any comparison between SMPD2 and SMPD4 do not seem very relevant.

Response: We thank for the concerning about this issue. According to the helpful suggestions, we have deleted the comparisons between hSMPD2 and ASMases, and mainly focused on the comparisons with SMPD3 and Isc1p in the comparison sections of the revised manuscript.

4. All the point mutations appear to be characterized at single concentration of sphingomyelin,

which limits the mechanistic insight of these residues in catalysis. To better understand the role of these residues (sphingomyelin affinity vs. catalysis rate) it would be useful (if the authors' assay is compatible) to determine hSMPD2 activity at different molar ratios of sphingomyelin to Triton X-100 to see if the point mutations affect apparent K_M or k_{cat} .

Response: We appreciate your comments. We have probed the enzymatic kinetic parameters of hSMPD2 with the substrate SM and lysoPAF. The representative point mutations showed that the I19A mutant increased the K_M value of hSMPD2 by catalyzing sphingomyelin, whereas the E49A mutant could not be fitted with an appropriate K_M value due to its low catalytic rate.

5. Lines 277-279: The statement “whereas hSMPD3 and ASMases have been proposed to adopt a similar dimeric structure through the N-terminal juxtamembrane saposin domains” should be reworded. ASMase has a saposin domain, hSMPD3 has a juxtamembrane region. I’m not aware of evidence for dimer formation in hSMPD3. Is there a reference for this?

Response: We appreciate your comments and we have corrected this error in the revised manuscript. In addition, we previously purified the overexpressed hSMPD3 protein and preliminary cryo-electron microscopy results showed that hSMPD3 adopts a lantern-shaped dimeric structure. For clarity, we removed this description in the revised manuscript.

6. Lines 280-281: The statement “The second deviation is that the D-K loop domain identified in hSMPD2 is not conserved among hSMPD families” should be reworded and/or put into context. The DK-loop is conserved in SMPD3, SMPD5, bacterial nSMases, and yeast Isc1p; it is only not conserved in SMPD4, which belongs to a different enzyme family. The sequence alignment included shows an alignment between SMPD2,3,4 which doesn’t provide much information since SMPD4 has a different protein fold and sequence homology is not relevant.

Response: We thank for the concerning about this issue. We have now corrected the sequence alignment from the HHpred web-server by combining the secondary structure prediction and the reported nSMases structures. As illustrated in our revised sequence alignment, residues Asp and Lys are conserved from human to bacteria, signifying a pivotal role in catalysis process. In addition, we have we have corrected the structural comparison section and mainly focused on the comparisons between SMPD2, SMPD3 and Isc1p.

7. Line 153: “In addition to nSMase3” should be “With the exception of nSMase3”. Also consider adding a citation for this if relevant.

Response: We have corrected this and added a citation (PMID: 16517606) in the revised manuscript.

8. Lines 144-148: The authors state: “hSMPD2 with Q91A, E92A, L356A, W367A, or T359A point mutations each had similar sphingomyelin hydrolysis activity compared to wild type hSMPD2 (Fig. 2d and Supplementary Fig. 5a), indicating that these residues predominantly participate in protein dimerization rather than in substrate hydrolysis”. I’m assuming that mutation of these residues does not eliminate dimer formation? Are the authors implying that dimer formation is not important for hSMPD2 activity?

Response: We thank for the comments and we have corrected the inappropriate description in the revised manuscript. Seeking experimental validation for dimerization of the residues, in vitro pull-down assays were performed with extracts from cells co-expressing Strep-tagged hSMPD2 and Flag-tagged hSMPD2 by using Strep-tagged proteins as “bait” protein.

The expression level of Q91A, L356A, T359A, and W367A are comparable to WT protein, whereas E92A mutants largely reduced the protein expression level compared with the wildtype, resulting in a decreased dimer stability. Multiple unbiased MD simulations (5 x 50 ns) after the equilibration phase indicates that E92 and K168 could form a stable salt bridge throughout the MD simulations, further verifying the essential role of this salt bridge interaction in dimer formation

According to the reviewer’s suggestions and our revised experiments, E92-K168 salt bridge interaction is important in stabilizing the dimeric architecture, whereas the other residues are simply in contact.

9. Lines 157-159: “Residue N15 is conserved in all nSMases and in their homolog in *Saccharomyces cerevisiae*, Isc1p, whereas residue E49 is only conserved among hSMPD2, hSMPD4, and Isc1p”. Are the authors sure that this glutamate residue is not conserved in hSMPD3? I would suspect it is not conserved in SMPD4 because that is a different protein fold, but it would be very surprising if SMPD3 does not conserve a key residue for Mg coordination.

Response: We thank for the concerning about this issue. We apologize for the inappropriate sequence alignment that leading to misleading conclusions. We have now corrected the sequence alignment from the HHpred web-server by combining the secondary structure prediction and the reported nSMases structures, which further indicating that the E49 and N15 from hSMPD2 are conserved within the nSMases (hSMPD3, MA-nSMase, Sc-ISC1, Li-smc1, Bc-SMase, Sa_Beta-toxin).

10. Line 181: consider rewording “the winning pose”. This phrasing does not make clear why this pose was chosen. There must be a logical reason why it was selected and by explicitly stating the reason, it would make clear why it was the binding pose most consistent with catalysis.

Response: We have re-organized the description in the revised manuscript as “The pose with a minimum distance (2.1Å) between the magnesium ion and the non-bridging oxygen atom of the phosphate group of sphingomyelin among 600 docking conformations was selected as the possible binding mode”

11. The I19A mutation appears to retain ~20% activity, not 5% (as shown in Fig. 3e). Is this a typo?

Response: We thank for the comments and we have corrected this error in the revised manuscript.

12. Lines 287-288: The statement “but such a conformation change in the saposin domain was not observed in the current hSMPD2 model” implies that SMPD2 has a saposin domain, which it does not. SMPD1 (acid sphingomyelinase) has a saposin domain but not SMPD2.

Response: We thank for the comments and we have corrected the misleading description in the revised manuscript.

REVIEWER COMMENTS

Reviewer #1 (Remarks to the Author):

The authors have responded well to my comments and I have no further concerns.

Minor corrections:

Lines 123, 303: "sequence *identity*"

Line 140/141 "four *acyl chains*"

Lines 237, 344 "the *amine* group of K116"

Reviewer #2 (Remarks to the Author):

The authors have improved the manuscript and they now provide some additional results. However, some points have not been adequately addressed:

Major points

1. The function as a sphingomyelinase in cells.

Some of the studies they cite on cellular biochemical activities of SMPD2 are not properly cited or evaluated;

- Jafferzou et al (ref19) don't refer specifically to SMPD2 as this study was developed prior to the molecular identification of SMPD2 and SMPD3;
- Yabu et al (17), they don't show the sequence of their siRNA so it is not clear what they were studying.
- Tonnetti et al (ref 18) show that both FB and knock down of SMPD2 inhibit ceramide production so it is not clear who comes first ceramide synthases or SMPD2).
- The study by Choezom & Gross ((25) actually implicates SMPD3 not SMPD2.
- Xing et al (26) report on using GW4869 to inhibit the changes in cells. However, this inhibitor does not act on nSMase1 but acts on nSMase2.

Therefore, with only a couple of papers claiming cellular sphingomyelinase activity for SMPD2 vs dozens on SMPD3, the authors should be more circumspect on this issue. Also, this reinforces the needs to do more work (e.g. modeling) with lysoPAF.

The localization of SM in the ER raises serious questions as SM is synthesized in the Golgi and not appreciated to be in the ER. Since the bodipy label is in the fatty acyl group, the fluorescece could be in any of a number of molecules (e.g. ceramide etc). Also the microscopy images are not convincing. Where are the nuclei? Where is the plasma membrane?

This reviewer does not find this experiment meaningful.

2. The authors have not clearly addressed whether they believe the protein traverses the membrane (trans membrane) or not.

Specific Points

1. Ok
2. These are supportive results yet they don't rule out effects of mutations on folding of the protein etc. some caution in making these statements should be applied.
3. Ok

New point

Introduction: the reference to Dolma et al should be to Choezom & Gross.

Reviewer #3 (Remarks to the Author):

The authors have significantly improved the manuscript through revision and all major comments have been addressed. There remains two minor typos to fix, and one suggestion for authors to consider.

1. line 165- the manuscript text was not changed as described in the rebuttal. "In addition to nSMase3" should be replaced with "With the exception of nSMase3"

2. line 182- the manuscript text was not changed as described in the rebuttal. "Michael et al" should be "Airola et al"

suggestion- lines 225-238 and Fig. 4b. Does D111 interact with K116 during simulations? If so, could the authors mention role of D111 in catalysis? e.g. interacting with K116 and any role they propose?

REVIEWER COMMENTS

Reviewer #1 (Remarks to the Author):

The authors have responded well to my comments and I have no further concerns.

Minor corrections:

Lines 123, 303: “sequence *identity*”

Line 140/141 “four *acyl chains*”

Lines 237, 344 “the *amine* group of K116”

Response: We thank for the comments and we have corrected the descriptions in the revised manuscript as suggested.

Reviewer #2 (Remarks to the Author):

The authors have improved the manuscript and they now provide some additional results. However, some points have not been adequately addressed:

Major points

1. The function as a sphingomyelinase in cells.

Some of the studies they cite on cellular biochemical activities of SMPD2 are not properly cited or evaluated;

- Jafferzou et al (ref19) don't refer specifically to SMPD2 as this study was developed prior to the molecular identification of SMPD2 and SMPD3;
- Yabu et al (17), they don't show the sequence of their siRNA so it is not clear what they were studying.
- Tonnetti et al (ref 18) show that both FB and knock down of SMPD2 inhibit ceramide production so it is not clear who comes first ceramide synthases or SMPD2).
- The study by Choezom & Gross ((25) actually implicates SMPD3 not SMPD2.
- Xing et al (26) report on using GW4869 to inhibit the changes in cells. However, this inhibitor does not act on nSMase1 but acts on nSMase2.

Therefore, with only a couple of papers claiming cellular sphingomyelinase activity for SMPD2 vs dozens on SMPD3, the authors should be more circumspect on this issue. Also, this reinforces the needs to do more work (e.g. modeling) with lysoPAF.

Response: Thanks for the valuable comments and suggestions. First, we have reorganized the description of the cellular function in the revised manuscript with more circumspect (line 58-83). Second, we have conducted a parallel comparison study using the Lyso-PAF as substrate via QM/MM US simulations. The free energy profile for Lyso-PAF along our proposed catalytic pathway demonstrated a higher free energy barrier compared with sphingomyelin and the hydrolysis reaction for Lyso-PAF is an

endergonic process with $G=2.34$ kcal/mol. According to our calculated free energy profiles, both the catalytic reaction would occur at room temperature although the Lyso-PAF exhibit a higher activation barrier. This result was added in Supplementary Fig.11.

Note: Free energy profiles for our proposed catalytic mechanism using SM (green line) and Lyso-PAF (purple line) as substrate, respectively

The localization of SM in the ER raises serious questions as SM is synthesized in the Golgi and not appreciated to be in the ER. Since the bodipy label is in the fatty acyl group, the fluorescence could be in any of a number of molecules (e.g. ceramide etc). Also the microscopy images are not convincing. Where are the nuclei? Where is the plasma membrane?

This reviewer does not find this experiment meaningful.

Response: We thank for the helpful comments. Given that the bodipy label is in the fatty acyl group and we could not distinguish the exact molecular forms of the cellular fluorescence, we have removed this inappropriate description in the revised manuscript.

2. The authors have not clearly addressed whether they believe the protein traverses the membrane (trans membrane) or not.

Response: Thanks for your valuable comments. The current hSMPD2 is a transmembrane protein. In the above figure, panel a shows that the transmembrane helices are surrounded by lipids *in vivo* or detergents *in vitro*. In addition, panel b shows the hydrophobic properties of the transmembrane helices. In the revised manuscript, including lines 32, 115, 118, and 310, we use "transmembrane helices" to clarify that hSMPD2 could trans membrane.

Specific Points

1. Ok
2. These are supportive results yet they don't rule out effects of mutations on folding of the protein etc. some caution in making these statements should be applied.

Response: We thank for the helpful comments and we have made caution with this point as suggested. Please see line 162 in the revised manuscript and now it reads "indicating that mutations in these residues might affect hSMPD2 activity."

3. Ok

New point

Introduction: the reference to Dolma et al should be to Choezom & Gross.

Response: We thank for the comments and we have corrected the descriptions in the revised manuscript as suggested.

Reviewer #3 (Remarks to the Author):

The authors have significantly improved the manuscript through revision and all major comments have been addressed. There remains two minor typos to fix, and one suggestion for authors to consider.

1. line 165- the manuscript text was not changed as described in the rebuttal. "In

addition to nSMase3" should be replaced with "With the exception of nSMase3"

Response: We thank for the comments and we have corrected the descriptions in the revised manuscript as suggested.

2. line 182- the manuscript text was not changed as described in the rebuttal. "Michael et al" should be "Airola et al"

Response: We thank for the comments and we have corrected the descriptions in the revised manuscript as suggested.

suggestion- lines 225-238 and Fig. 4b. Does D111 interact with K116 during simulations? If so, could the authors mention role of D111 in catalysis? e.g. interacting with K116 and any role they propose?

Response: We thank for the helpful suggestions. Our unbiased MD simulations indicated that a stable salt-bridge was formed between D111 and K116 during five independent MD trajectories. The same role of D111 was also observed in the catalysis process during the QM/MM simulations, see supplementary fig 9 in the revised manuscript.

REVIEWER COMMENTS

Reviewer #1 (Remarks to the Author):

The authors have responded well to my comments and I have no further concerns.

Minor corrections:

Lines 123, 303: “sequence *identity*”

Line 140/141 “four *acyl chains*”

Lines 237, 344 “the *amine* group of K116”

Response: We thank for the comments and we have corrected the descriptions in the revised manuscript as suggested.

Reviewer #2 (Remarks to the Author):

The authors have improved the manuscript and they now provide some additional results. However, some points have not been adequately addressed:

Major points

1. The function as a sphingomyelinase in cells.

Some of the studies they cite on cellular biochemical activities of SMPD2 are not properly cited or evaluated;

- Jafferzou et al (ref19) don't refer specifically to SMPD2 as this study was developed prior to the molecular identification of SMPD2 and SMPD3;
- Yabu et al (17), they don't show the sequence of their siRNA so it is not clear what they were studying.
- Tonnetti et al (ref 18) show that both FB and knock down of SMPD2 inhibit ceramide production so it is not clear who comes first ceramide synthases or SMPD2).
- The study by Choezom & Gross ((25) actually implicates SMPD3 not SMPD2.
- Xing et al (26) report on using GW4869 to inhibit the changes in cells. However, this inhibitor does not act on nSMase1 but acts on nSMase2.

Therefore, with only a couple of papers claiming cellular sphingomyelinase activity for SMPD2 vs dozens on SMPD3, the authors should be more circumspect on this issue. Also, this reinforces the needs to do more work (e.g. modeling) with lysoPAF.

Response: Thanks for the valuable comments and suggestions. First, we have reorganized the description of the cellular function in the revised manuscript with more circumspect (line 60-80). Second, we have conducted a parallel comparison study using the Lyso-PAF as substrate via QM/MM US simulations. The free energy profile for Lyso-PAF along our proposed catalytic pathway demonstrated a higher free energy barrier compared with sphingomyelin and the hydrolysis reaction for Lyso-PAF is an endergonic

process with $G=2.34$ kcal/mol. According to our calculated free energy profiles, both the catalytic reaction would occur at room temperature although the Lyso-PAF exhibit a higher activation barrier. This result was added in Supplementary Fig.11 and 12.

Note: Free energy profiles for our proposed catalytic mechanism using SM (green line) and Lyso-PAF (purple line) as substrate, respectively

The localization of SM in the ER raises serious questions as SM is synthesized in the Golgi and not appreciated to be in the ER. Since the bodipy label is in the fatty acyl group, the fluorescence could be in any of a number of molecules (e.g. ceramide etc). Also the microscopy images are not convincing. Where are the nuclei? Where is the plasma membrane?

This reviewer does not find this experiment meaningful.

Response: We thank for the valuable comments. Fluorescently labeled sphingomyelin molecular probes have been developed and applied in several studies (PMID: 9668349; 11001561; 9864149). In the revised manuscript, we show the results of co-localization of hSMPD2 with labeled SM by presenting the ER, Golgi, nucleus, and plasma membrane separately (Supplementary Fig 1). The overexpressed hSMPD2 was mainly localized to the ER and partially to the Golgi and the plasma membrane. The sphingomyelin is reported to be enriched in the plasma membrane, the endocytic recycling compartment, and the trans Golgi network (PMID: 23684760). By incubating the BODIPY labeled SM for 1h, we observed the fluorescence co-localization between SM and overexpressed hSMPD2 on the Golgi, plasma membrane, and ER. Because the current hSMPD2 is overexpressed and lipids are added from the culture medium and undergo membrane lipid fusion and endocytic recirculation, this co-localization assay is not fully representative of the characteristics of endogenous hSMPD2 and SM. We have mentioned this in the discussion section (line 370-374) that our current study provides catalytic properties of sphingomyelin and lyso-PAF from the structure side and lays the foundation for more in-depth further investigations.

2. The authors have not clearly addressed whether they believe the protein traverses the membrane (trans membrane) or not.

Response: Thanks for your valuable comments. The current hSMPD2 is a transmembrane protein. In the above figure, panel a shows that the transmembrane helices are surrounded by lipids in vivo or detergents in vitro. In addition, panel b shows the hydrophobic properties of the transmembrane helices. In the revised manuscript, including lines 32, 122, 125, and 317, we use "transmembrane helices" to clarify that hSMPD2 could trans membrane.

Specific Points

1. Ok
2. These are supportive results yet they don't rule out effects of mutations on folding of the protein etc. some caution in making these statements should be applied.

Response: We thank for the helpful comments and we have made caution with this point as suggested. Please see line 169 in the revised manuscript and now it reads "indicating that mutations in these residues might affect hSMPD2 activity."

3. Ok

New point

Introduction: the reference to Dolma et al should be to Choezom & Gross.

Response: We thank for the comments and we have corrected the descriptions in the revised manuscript as suggested.

Reviewer #3 (Remarks to the Author):

The authors have significantly improved the manuscript through revision and all major comments have been addressed. There remains two minor typos to fix, and one suggestion for authors to consider.

1. line 165- the manuscript text was not changed as described in the rebuttal. "In addition to nSMase3" should be replaced with "With the exception of nSMase3"

Response: We thank for the comments and we have corrected the descriptions in the revised manuscript as suggested.

2. line 182- the manuscript text was not changed as described in the rebuttal. "Michael et al" should be "Airola et al"

Response: We thank for the comments and we have corrected the descriptions in the revised manuscript as suggested.

suggestion- lines 225-238 and Fig. 4b. Does D111 interact with K116 during simulations? If so, could the authors mention role of D111 in catalysis? e.g. interacting with K116 and any role they propose?

Response: We thank for the helpful suggestions. Our unbiased MD simulations indicated that a stable salt-bridge was formed between D111 and K116 during five independent MD trajectories. The same role of D111 was also observed in the catalysis process during the QM/MM simulations, see supplementary fig 10 in the revised manuscript.

REVIEWERS' COMMENTS

Reviewer #2 (Remarks to the Author):

The authors have addressed most of my comments. The sole remaining issue is the ER localization. the supplementary figure is simply not convincing. while the localization of SMPD2 appears reticular, the authors have not used a usual ER marker. instead they used GS28 which is predominantly a Golgi protein! this has to be rectified. indeed the patten of the lipid staining looks clearly Golgi.

REVIEWER COMMENTS

Reviewer #2 (Remarks to the Author):

The authors have addressed most of my comments. The sole remaining issue is the ER localization. the supplementary figure is simply not convincing. while the localization of SMPD2 appears reticular, the authors have not used a usual ER marker. instead they used GS28 which is predominantly a Golgi protein! this has to be rectified. indeed the patten of the lipid staining looks clearly Golgi.

Response: Thanks for the valuable comments and suggestions. We have performed the localization assay by using the Sec61B antibody to trace the ER location. This has been rectified in the revised supplementary figure 1.